# Linear Antenna Array Sectorized Beam Scanning Approaches Using Element Position Perturbation in the Azimuth Plane

**DOI:** 10.3390/s23146557

**Published:** 2023-07-20

**Authors:** Safaa I. Abd Elrahman, Ahmed M. Elkhawaga, Amr H. Hussein, Abd Elhameed A. Shaalan

**Affiliations:** 1Electronics and Communication Engineering Department, Faculty of Engineering, Zagazig University, Zagazig 44519, Egypt; safaa9ibrahim@gmail.com; 2Electronics and Electrical Communications Engineering Department, Faculty of Engineering, Tanta University, Tanta 31527, Egypt; 3Electronics and Electrical Communications Engineering Department, Faculty of Engineering, Horus University Egypt, New Damietta 34518, Egypt

**Keywords:** array thinning, beam scanning approach (BSA), element position perturbations (EPP), linear antenna array (LAA), side lobe level (SLL), half power beamwidth (HPBW), genetic algorithm (GA), planar antenna array (PAA)

## Abstract

In this paper, two sector beam scanning approaches (BSAs) based on element position perturbations (EPPs) in the azimuth plane are introduced. In EPP-BSA, the elements’ excitations are kept constant and the elements’ positions in the direction normal to the array line are changed according to a predetermined EPP pattern. The magnitude and repetition rate of the selected EPP pattern determines the steering angle of the main beam. However, EPP-BSA results in a wide scanning range with a significant increase in the side lobe level (SLL). To mitigate this drawback, a reduction in the SLL of the array pattern is firstly performed using the single convolution/genetic algorithm (SC/GA) technique and then perturbing the elements’ positions in the azimuth plane. This combination between SLL reduction and EPP-BSA (SLL/EPP-BSA) results in a smaller scanning range with a relatively constant half power beamwidth (HPBW) and a much lower SLL. In addition, keeping the synthesized excitation coefficients constant without adding progressive phase shifters facilitates the manufacturing process and reduces the cost of the feeding network. Furthermore, a planar antenna array thinning approach is proposed to realize the EPP-BSA. The results are realized using the computer simulation technology (CST) microwave studio software package, which provides users with an optimized modeling environment and results in realizable and realistic designs.

## 1. Introduction

In both military and civilian applications, such as wireless communication systems, wireless power transmission systems, and radar systems, phased antenna arrays have various benefits including flexible beam scanning, high tracking accuracy, and widespread use [1]. Flexible beam-scanning antenna design is currently a hot topic of research because of the emergence of contemporary applications that need antenna arrays with variable scanning capabilities [2]. Automotive radars, weather observations, aviation surveillance [3], and satellite communication [4] are just a few of these numerous applications.

In [5], a $\left(2\times 3\right)$ phased antenna array is developed using a rectangular microstrip patch antenna on a FR-4 dielectric substrate. The elements are excited with constant amplitude and different phases to steer the beam within a scanning range of 49° (from −25° to +24°). In [6], four leaf-shaped bow-tie slot antennas in a linear array are used for beam scanning within a scanning range from 9° to 13°. In [7], the two-element antenna array is fed through two transmission lines that are inclined at an angle from a common base line. The main beam can be directed from one antenna’s normal to the other antenna’s normal by adjusting the excitations of the two antennas. The scan angle depended on changing the excitation coefficient in every scan angle or changing the tilt angle. In [8], a broadband reconfigurable antenna that can guide the main beam in three distinct directions has been introduced. Besides the main radiating patch, three parasitic tiny patches are included to control the main beam steering. Three beam-steering orientations of the radiation pattern are offered by PIN diode switching. In [9], the same concept of switched parasitic patches at 60 GHz for WLAN applications has been introduced for main beam steering. In [10], a sparse substrate integrated waveguide (SIW) slot antenna array has been introduced. The genetic algorithm was utilized to optimize the position of each element over the aperture of $4.5{\;\lambda }_{o}$, where $\lambda_{o}$ is the free space wavelength. The sparse array consists of array elements, phase shifters, and power dividers on a substrate integrated with an SIW. This sparse antenna can scan the beam in the range of ±15°. In [11], a millimeter-wave (MMW) multi-folded frequency-dependent progressive phase shifter based on a substrate integrated waveguide (SIW) has been introduced. By combining the antenna element and the phase shifter together, the array can achieve a wide scanning range from −18° to 32°.

On the other hand, array thinning has been utilized for large array synthesis to achieve desired radiation patterns with a minimum array size and cost [12,13,14,15,16]. In [12], a numerical stochastic optimization technique that is known as multiagent genetic algorithm (MAGA) has been used for planar array thinning and synthesis for both low SLL and high directivity. In [13], discrete particle swarm optimization (DPSO) has been used for large array thinning. The peak side lobe level (PSLL) performance of the algorithm is verified through several representative examples, demonstrating its effectiveness and robustness.

In [14], the binary genetic algorithm (BGA) has been used for thinned planar array design, where only the outer sub-planar array elements with small amplitude excitations are optimized for thinning. The BGA has been used to remove a number of useless antenna elements while keeping the inner sub-planar elements with high amplitude excitations unchanged. In [15], a GA in combination with array thinning and tapering techniques has been introduced to minimize the SLL. Thinning is applied to idle some elements of the array, and tapering is used to achieve a superior SLL. In [16], a hybrid approach combining difference sets (DS) and discrete Fourier transform (DFT) techniques has been introduced for the synthesis of thinned planar antenna arrays. DS is used to establish pattern synthesis rules, while DFT is employed to find the nonuniform excitations of a planar antenna array. The proposed method extracts excitations using cyclic shifts from a suitable DS sequence to achieve antenna array thinning with a reduced PSLL.

In this paper, two beamforming approaches for sectorized beam scanning of linear antenna arrays based on the element position perturbation (EPP) technique denoted as EPP-BSA and SLL/EPP-BSA are introduced. In both techniques, the elements’ excitations are kept constant while the positions of the elements are perturbed in the azimuth plane according to a predefined perturbation pattern. In this work, the cosine wave pattern is utilized for this purpose. The proposed EPP-BSA provides wider scanning ranges than the proposed SLL/EPP-BSA but with much higher SLLs at the endpoints of the scanning range. In this regard, the proposed SLL/EPP-BSA may be useful in interference-sensitive applications that require small scanning sectors. The proposed techniques are based on modifying the EPP pattern rather than using phase shifters while maintaining the elements excitations. This simplifies the production process and lowers the cost of the feeding network. Furthermore, the proposed technique eliminates interference and coupling between the main beam and the back lobe at any scanning angle by maintaining a 180 spacing angle between them. Furthermore, a proposed PAA thinning approach is proposed to realize the proposed EPP-BSA approach. It allows for precise array thinning to select the desired active “ON” antenna elements and turn off the other elements to configure the designed EPP pattern to achieve a specific scan angle with the desired radiation properties. The results are realized using the computer simulation technology (CST) microwave studio software package, which is highly matched to the MATLAB simulations.

The main contributions of this work can be summarized as follows:The proposed techniques provide continuous beam steering with high accuracy rather than switched beam steering or discrete beam steering techniques.The required beam steering is simply achieved by only adjusting the amplitude and repetition rate of the utilized perturbation pattern that is fully controlled.The elements’ excitations (magnitude and phase) are kept constant throughout the entire steering range. That completely avoids the design of complex and expensive feeding networks. On the other hand, the existing beam steering techniques are mainly based on utilizing n-bit phase shifters that are complex and expensive. Also, the accuracy of main beam steering depends on the phase quantization errors of the utilized phase shifters arising from the limited number of bits per phase shifter.Although the EPP has been employed for pattern nulling or side lobe cancellation, it has been utilized in a novel manner for achieving continuous beam steering techniques that can be applicable for beam steering of millimeter wave patch antenna arrays as described in the paper.With an HPBW decrease of around 3° at the extremities of the
steering range, the first technique, EPP-BSA, offers a wide steering range with a width of 102.72°. While at the extremes of the steering range, it experiences a high SLL of −2.668 dB.The second technique, SLL/EPP-BSA, provides a steering range of width 61.46° with fixed HPBW and SLL<−10 dB over the entire steering range.A proposed PAA thinning approach is introduced to realize the proposed EPP-BSA technique. It allows for precise array thinning to select the desired active “ON” antenna elements and turn off “OFF” the other elements to configure the designed EPP pattern to steer the main beam in a specific direction with the desired radiation properties.

## 2. Proposed EPP-Based Beam Scanning Approaches

This section introduces two array beam scanning approaches that employ element position perturbation (EPP) in the azimuth plane. In these techniques, the array elements are distributed using well-known waveform patterns such as sine, cosine, and triangle waves. In the first approach, the original excitations of the elements are kept constant. While the amplitude and repetition rate of the chosen waveform pattern along the array line define the element position and the main beam’s steering angle. So, it can be denoted as the EPP-based scanning approach (EPP-BSA). However, the EPP-BSA results in a wide scanning range with a significant increase in the side lobe level (SLL).

To mitigate the SLL problem of the EPP-BSA, a second array beam scanning approach is introduced. In this approach, SLL reduction of the array pattern is firstly performed using the single convolution/genetic algorithm (SC/GA) technique introduced in [17] and then performance-optimized EPP in the azimuth plane. This combination between SLL reduction and EPP (SLL/EPP) results in a limited scanning range but with a relatively constant HPBW and minimal variations in the synthesized SLL. The proposed SLL/EPP-based beam scanning approach can be denoted as the SLL/EPP-BSA.

### 2.1. Proposed EPP-BSA

To derive a closed form expression of the synthesized array factor applying EPP, we considered a linear antenna array (LAA) whose elements distributions are shown in Figure 1 before and after EPP.

Consider a uniform LAA consisting of N antenna elements with uniform element spacing d as shown in Figure 2, its array factor $AF\left(\theta \right)$ is given by [18]:(1)$$AF\left(\theta \right)=\sum_{n=1}^{N}a_{n\;}e^{jk\left(n-1\right)d\cos\theta }$$
where $a_{n}$ is the excitation coefficient of the $n^{th}$ antenna elements, k=2π/λ is the wave number, and λ is the wavelength.

If the positions of the N antenna elements of the linear antenna array are perturbed along the *Y*-axis as shown in Figure 2, the array factor of Equation (1) should be modified. The total electric field $E_{t}$ at the far field point is the superposition of the individual electric fields arising from each antenna element such that:(2)$$E_{t}=E_{1+}E_{2}+\cdots \cdots \cdots +E_{N}$$
where
(3)$$E_{1}=a_{1}\frac{E_{0}}{r_{1}}e^{-jkr_{1}}$$
(4)$$E_{2}=a_{2}\frac{E_{0}}{r_{2}}e^{-jkr_{2}}$$
(5)$$E_{N}=a_{N}\frac{E_{0}}{r_{N-1}}e^{-jkr_{N-1}}$$

Then, the total field can be expressed as:(6)$$E_{t}=\left(\frac{E_{0}}{r_{1}}a_{1}e^{-jkr_{1}}+\frac{E_{0}}{r_{2}}a_{2}e^{-jkr_{2}}+\cdots +a_{N}\frac{E_{0}}{r_{N-1}}e^{-jkr_{N-1}}\right)$$
where $r_{n}$, n=1,2, 3, …., N is the distances from the $n^{th}$ antenna element to the far field point. For far field approximation we use the following assumptions:(7)$$\frac{1}{r_{1}}=\frac{1}{r_{2}}=\frac{1}{r_{3}}=\cdots =\frac{1}{r_{N}}=\frac{1}{r}$$
(8)$$r_{1}∥r_{2}∥r_{3}∥\cdots ∥r_{N}$$
where r is the distance from the origin or reference point to the far field point and ∥ is the parallel operator. And, θ is the angle between the r vector and the *Z*-axis. d is the distance between successive elements on the *Z*-axis, $d_{yn}$ is the distance of the $n^{th}$ element in the Y-direction, $d_{n}$ is distance between the $n^{th}$ element and origin, and $\theta_{n}$ is the angle between $d_{n}$ and the *Z*-axis as shown in Figure 3.

Based on Figure 3, the angle $\theta_{n}$ can be calculated as follows:(9)$$\sin\theta_{n}=\frac{d_{yn}}{d_{n}}$$
(10)$$\cos\theta_{n}=\frac{\left(n-1\right)\;d}{d_{n}}$$

To determine $r_{1}$ for the first antenna element consider the array geometry shown in Figure 4. In this case, $r_{1}$ can be expressed as:(11)$$r_{1}=r-d_{y1}\sin\theta$$

On the other hand, $r_{2}$ of the second antenna element can be determined as shown in Figure 5 such that: $r_{2}=r-d_{2}\cos\left(\theta_{2}-\theta \right)$
(12)$$r_{2}=r-d_{2}\cos\theta \cos\theta_{2}-d_{2}\sin\theta \sin\theta_{2}$$

Substituting $\cos\theta_{2}=\frac{d}{d_{2}}$ and $\sin\theta_{2}=\frac{{d_{y}}_{2}}{d_{2}}$ into Equation (12) then
(13)$$r_{2}=r-d_{2}\cos\theta \times \frac{d}{d_{2}}-d_{2}\sin\theta \times \frac{{d_{y}}_{2}}{d_{2}}\;r_{2}=r-d\cos\theta -{d_{y}}_{2}\sin\theta$$

By the same way, $r_{3}$ of the third antenna element can be determined as:(14)$$r_{3}=r-2d\cos\theta -{d_{y}}_{3}\sin\theta$$

In general, the $r_{n}$ of the $n^{th}$ antenna element can be expressed as:(15)$$r_{n}=r-\left(n-1\right)d\cos\theta -{d_{y}}_{n}\sin\theta$$

Substituting the far field approximations in Equation (6), the total electric field $E_{t}$ can be rewritten as:(16)$$E_{t}=\frac{E_{0}}{r}e^{-jkr}\left(a_{1}e^{jk{d_{y}}_{1}\sin\theta }+a_{2}e^{jkd\cos\theta +jk{d_{y}}_{2}\sin\theta }+\cdots +a_{N}e^{jk\left(N-1\right)d\cos\theta +jk{d_{y}}_{N}\sin\theta }\right)$$

Then,
(17)$$E_{t}=\frac{E_{0}}{r}e^{-jkr}\times AF\left(\theta \right)$$

Accordingly, the modified array factor can be expressed as:(18)$$AF\left(\theta \right)=a_{1}e^{jk{d_{y}}_{1}\sin\theta }+a_{2}e^{jkd\cos\theta +jk{d_{y}}_{2}\sin\theta }+\cdots +a_{N}e^{jk\left(N-1\right)d\cos\theta +jk{d_{y}}_{N}\sin\theta }$$

In other words, it can be written as:(19)$$AF\left(\theta \right)=\sum_{n=1}^{N}a_{n}e^{\left(jk\left(n-1\right)d\cos\theta +jkd_{y_{n}}\sin\theta \right)}$$

It is clear that $AF\left(\theta \right)$ is a function of the element position perturbation,$\;{\begin{array}{c} d \end{array}}_{y_{n}}=\left[\;d_{y1}\;{\;d}_{y2}\ldots \;d_{y_{N}}\right]$ in the *Y*-direction.

#### 2.1.1. Simulation Results of the Proposed EPP-BSA

In this section, several simulations are carried out to verify the proposed approach and give a recommendation for the best case. Consider a broadside uniform linear antenna array (ULAA) consisting of N=8 elements with uniform element spacing d=λ/2 whose elements are arranged along the *Z*-axis. It is required to scan the array main beam to the right and left of the broadside direction. This is achieved by performing EPP in the *Y*-direction following a known waveform pattern. In this case, the cosine waveform is utilized for the EPP where the cosine wave amplitude and period are expressed in terms of the operating wavelength λ. The half period H of the cosine wave along the *Z*-axis is chosen related to the array length $L_{h}=\left(N-1\right)\times d=3.5\;\lambda$.

In this case, the element position in the Y-direction along the array line (*Z*-direction) is given by:(20)$${\begin{array}{c} d \end{array}}_{y_{n}}=\pm A_{c}\times cos\left(\frac{z\times \pi }{H}\right)$$
where $A_{c}$ is the amplitude of the cosine wave, z is the element position on the *Z*-axis, and H is the half period of cosine that is chosen to be $H\geq L_{h}$. Both the amplitude of the cosine wave $A_{c}$ and the half period H control the scanning angle of the array main beam.

The beam scanning is performed at different compression ratios R that are defined as:(21)$$R=\frac{L_{h}}{H},\;\;H\geq L_{h}$$

It is noticed that as H increases more than $L_{h}$, the array size in the Y-direction is reduced and becomes compact, so R is denoted as the compression ratio. Consequently, there are different test cases for beam scanning for different values of compression ratio.


**Test case (1): **

R=1



In this case, the cosine wave half period is chosen to be equal to the array size, $H=L_{h}=3.5\;\lambda$, hence R=1. Accordingly, the EPPs of the array elements for $A_{c}$ changing from positive to negative and $A_{c}$ changing from negative to positive are shown in Figure 6a and Figure 6b, respectively.

When the cosine wave amplitude is changing from positive to negative and its magnitude is changing from $A_{c}=0$ to $A_{c}=0.19\;\lambda$, the array main beam is scanned from 0° to 7.01° with HPBW and the SLL is less than −10 dB. However, as the cosine wave amplitude increases from than 0.19 λ to 1.5 λ, the scan angle is increased from 7.01° to 44.6°, the SLL is increased more than −10 dB, and the HPBW is decreased by 3°. However, as the cosine wave amplitude increases from than 1.5 λ to 2 λ, the scan angle is increased from 44.6° to 52.9°. But, the grating lobe appears, and the SLL is increased by more than −10 dB as listed in Table 1. The synthesized radiation patterns using the EPP technique are shown in Figure 7.

On the other hand, when the cosine wave amplitude is changing from negative to positive and its magnitude is changing from $A_{c}=0$ to $A_{c}=0.19\lambda$, the array main beam is scanned from 0° to −7.01° with a constant HPBW and the SLL is less than −10 dB. However, as the cosine wave amplitude increases from 0.19 λ to 1.5 λ, the scan angle is increased from −7.01° to −44.6°, the SLL is increased by more than −10 dB, and the HPBW is decreased by 3°. But, as the cosine wave amplitude increases from 1.5 λ to 2 λ, the scan angle is increased from −44.6° to −52.9° and the grating lobe appears with increased SLL as shown in Figure 8. The achieved maximum scanning range is $-52.91^{{}^{\circ}}{\leq \theta_{r}\leq 52.91}^{{}^{\circ}}$ around the broadside direction.

Unlike traditional progressive phase shift scanning, as the scanning angle increases away from the broadside direction, the SLL increases, and the HPBW is significantly increased many folds of its value in the broadside case as shown in Figure 7 and Figure 8. However, in the proposed EPP-BSA approach, as the amplitude of the cosine wave $A_{c}$ increases, the HPBW of the main beam decreases as the length or size of the array projection on the plane normal to the main beam direction $\theta_{o}$ is increased.


**Test case (2): **

R=0.875



In this case, the cosine wave half period is chosen to be greater than the array size, $H>L_{h}$ and equals H=4 λ, hence R=3.5 λ/4 λ=0.875. Accordingly, the EPPs of the array elements for $A_{c}$ changing from positive to negative and $A_{c}$ changing from negative to positive are shown in Figure 9a and Figure 9b, respectively.

When the cosine wave amplitude is changing from positive to negative and its magnitude is changing from $A_{c}=0$ to $A_{c}=0.25\;\lambda$, the array main beam is scanned from 0° to 8.56° with a constant HPBW and the SLL is less than −10 dB. However, as the cosine wave amplitude increases from 0.25 λ to 2 λ, the scan angle is increased from 8.56° to 51.36°, the SLL is increased by more than −10 dB, and the HPBW is decreased by 3.6° as listed in Table 2. The synthesized radiation patterns using the EPP technique are shown in Figure 10. It is clear that as the amplitude of the cosine waveform increases, the scanning angle increases away from the broadside direction, the HPBW decreases, and the SLL increases as shown in Figure 11.

On the other hand, when the cosine wave amplitude is changing from negative to positive and its magnitude is changing from $A_{c}=0$ to $A_{c}=0.25\;\lambda$, the array main beam is scanned from 0° to−8.56° with a constant HPBW and the SLL is less than −10 dB. However, as the cosine wave amplitude increases from 0.25 λ to 2 λ, the scan angle is increased from −8.56° to−51.36°, the SLL is increased by more than −10 dB, and the HPBW is decreased by 3.6° as shown in Figure 12. The achieved maximum scanning range is $-51.36{}^{\circ}\leq \theta_{r}\leq 51.36{}^{\circ}$ around the broadside direction.


**Test case (3): **

R=0.7778



In this case, the cosine wave half period is chosen to be greater than the array size, $H>L_{h}$ and equals H=4 λ, hence R=4.5 λ/4 λ=0.7778. Accordingly, the EPPs of the array elements for $A_{c}$ changing from positive to negative and $A_{c}$ changing from negative to positive are shown in Figure 13a and Figure 13b, respectively.

When the cosine wave amplitude is changing from positive to negative and its magnitude is changing from $A_{c}=0$ to $A_{c}=0.3\;\lambda$, the array main beam is scanned from 0° to 9.08° with a constant HPBW and the SLL is less than −10 dB. However, as the cosine wave amplitude increases from 0.3 λ to 2 λ, the scan angle is increased from 9.08° to 48.78°, the SLL is increased by more than −10 dB, and the HPBW is decreased by 1.38° as listed in Table 3. The synthesized radiation patterns using the EPP technique are shown in Figure 14. On the other hand, when the cosine wave amplitude is changing from negative to positive and its magnitude is changing from $A_{c}=0$ to $A_{c}=0.3\;\lambda$, the array main beam is scanned from 0° to−9.08° with a constant HPBW and the SLL is less than −10 dB. However, as the cosine wave amplitude increases from than 0.3 λ to 2 λ, the scan angle is increased from −9.08° to−48.78°, the SLL is increased by more than −10 dB, and the HPBW is decreased by 1.38° as shown in Figure 15. The achieved maximum scanning range is $-48.78{}^{\circ}\leq \theta_{r}\leq 48.78{}^{\circ}$ around the broadside direction.


**Test case (4): **

R=0.7



In this case, the cosine wave half period is chosen to be equal to H=5 λ, and the array length equals $L_{h}=3.5\;\lambda$, hence R=0.7. Accordingly, the EPPs of the array elements for $A_{c}$ changing from positive to negative and $A_{c}$ changing from negative to positive are shown in Figure 16a and Figure 16b, respectively.

When the cosine wave amplitude is changing from positive to negative and its magnitude is changing from $A_{c}=0$ to $A_{c}=0.39\;\lambda$, the array main beam is scanned from 0° to 10.68° with a constant HPBW and the SLL is less than −10 dB. However, as the cosine wave amplitude increases from 0.39 λ to 2 λ, the scan angle is increased from 10.68° to 45.69°, the SLL is increased by more than −10 dB, and the HPBW is decreased by 1.1° as listed in Table 4. The synthesized radiation patterns using the EPP technique are shown in Figure 17.

On the other hand, when the cosine wave amplitude is changing from negative to positive and its magnitude is changing from $A_{c}=0$ to $A_{c}=0.39\;\lambda$, the array main beam is scanned from 0° to−10.68° with a constant HPBW and the SLL is less than −10 dB. However, as the cosine wave amplitude increases from 0.39 λ to 2 λ, the scan angle is increased from −10.68° to−45.69°, the SLL is increased by more than −10 dB, and the HPBW is decreased by 1.1° as shown in Figure 18. The achieved maximum scanning range is $-45.69{}^{\circ}\leq \theta_{r}\leq 45.69{}^{\circ}$ around the broadside direction.


**Test case (5): **

R=0.5



In this case, the cosine wave half period is chosen to be equal to H=7 λ, and the array length equals $L_{h}=3.5\;\lambda$, hence R=0.5. Accordingly, the EPPs of the array elements for $A_{c}$ changing from positive to negative and $A_{c}$ changing from negative to positive are shown in Figure 19a and Figure 19b, respectively.

When the cosine wave amplitude is changing from positive to negative and its magnitude is changing from $A_{c}=0$ to $A_{c}=0.65\;\lambda$, the array main beam is scanned from 0°to 10.83 ° with a constant HPBW and the SLL is less than −10 dB. However, as the cosine wave amplitude increases from 0.65 λ to 2 λ, the scan angle is increased from $10.83^{{}^{\circ}}\;to\;31.25\;{}^{\circ}$, the SLL is increased by more than −10 dB, and the HPBW is decreased by 2.83° as listed in Table 5. The synthesized radiation patterns using the EPP technique are shown in Figure 20.

On the other hand, when the cosine wave amplitude is changing from negative to positive and its magnitude is changing from $A_{c}=0$ to $A_{c}=0.65\;\lambda$, the array main beam is scanned from 0° to−10.83° with a constant HPBW and the SLL is less than −10 dB. However, as the cosine wave amplitude increases from 0.65 λ to 2 λ , the scan angle is increased from −10.83° to−31.25°, the SLL is increased by more than −10 dB, and the HPBW is decreased by 2.83° as shown in Figure 21. The achieved maximum scanning range is $-31.25{}^{\circ}\leq \theta_{r}\leq 31.25{}^{\circ}$ around the broadside direction.

#### 2.1.2. Comparison between the Five Test Cases of Compression Ratio

The results of the aforementioned five test cases of the compression ratio R are summarized in Table 6. It is clear that for test cases 1, 2, and 3, as the cosine wave amplitude increases, the scanning angle increases, and the HPBW decreases. However, test case 2 at R=0.875, provides the largest decrease in the HPBW. On the other hand, for test cases 4 and 5, as the cosine wave amplitude increases, the scanning angle increases giving rise to a larger scanning range than in the previous test cases, and the HPBW increases. But, the null between the first side lobe and main beam is relatively high, resulting in stronger interference. As a consequence, we can conclude that test case 2 with *R* = 0.875 yields the best results in terms of HPBW and scanning range.

#### 2.1.3. CST Realization of the Proposed EPP-BSA

In this section, the proposed EPP-BSA is realistically validated for actual antenna elements rather than isotropic antennas. The antenna array is created using the CST microwave studio software package utilizing a dipole element, the dimensions of which, as well as the H-plane and E-plane patterns, are shown in Figure 22. The simulated scattering parameter (reflection coefficient) $\left|S_{11}\right|$ of the dipole antenna is illustrated in Figure 23, showing a resonance frequency $f_{o}=1\;GHz$.

Considering the recommended test case 2 with a compression ratio R=0.875, which yields the best results in terms of HPBW and scanning range, we created the synthesized eight-element antenna arrays applying the proposed EPP-BSA approach. The synthesized antenna arrays and the associated 3D radiation patterns are shown in Figure 24, while the polar plots of the synthesized radiation patterns using the CST software package are shown in Figure 25. The CST simulation results for $A_{c}$ changing from 0 to 2 λ are recorded in Table 7 compared to the MATLAB simulation results indicating the resultant scan angle, SLL, and HPBW. The comparison shows a high matching between the CST and MATLAB simulation results with minimal fractal changes because the CST acts as a real environment and considers the mutual coupling between antenna elements in the synthesized array structures. This demonstrates the ability of practical validation of the proposed EPP-BSA approach.

### 2.2. Proposed SLL/EPP-BSA

To mitigate the SLL problem of the EPP-BSA, the SLL/EPP-based beam scanning approach, which is denoted as SLL/EPP-BSA, is introduced. In this approach, SLL reduction of the array pattern is firstly performed using our single convolution/genetic algorithm (SC/GA) technique introduced in [13] and then performance-optimized EPP in the azimuth plane. This combination between the SLL reduction and the EPP (SLL/EPP) results in a wide scanning range from the broadside to the end-fire direction with a relatively constant HPBW and minimal variations in the synthesized SLL. In addition, keeping the synthesized excitation coefficients constant without adding progressive phase shifters facilitates the manufacturing process and reduces the cost of the feeding network. The SC/GA SLL reduction technique is utilized as it provides a twofold decrease in the SLL. Consider the ULAA configuration shown in Figure 1 whose array factor $AF\left(\theta \right)$ is defined by Equation (1), where $a_{n\;}$ is the excitation coefficient of the $n^{th}$ antenna element. The SC is used to determine the synthesized excitation coefficients, while the GA is utilized to determine the element spacing between the antenna array elements. The SC/GA technique can be summarized as follows:

For N-element ULAA, the N×1 excitation vector $A_{N\times 1}=\left[a_{1}\;a_{2}\;a_{3}\;a_{4}\ldots a_{N}\right]$ is convolved by itself such that the resultant 1D convolution vector $C_{\left(2N-1\right)\times 1}$ can be expressed as:(22)$$C_{\left(2N-1\right)\times 1}=A_{N\times 1}*A_{N\times 1}$$

However, the size of the resultant excitation vector $C_{I\times 1}=C_{\left(2N-1\right)\times 1}$ from the single convolution process is much larger than the size of the original excitation vector $A_{N\times 1}$. In order to synthesis the array factor ${AF}_{S}\left(\theta \right)$ using a reduced number of antenna elements, we divided the vector $C_{I\times 1}=C_{\left(2N-1\right)\times 1}$ into two vectors $C_{{\left(N\times 1\right)}_{O}}$ and $C_{\left(N-1\right)\times 1_{E}}$ that contain the odd and even excitations to implement the synthesized array factors ${AF}_{SO}\left(\theta \right)$ and ${AF}_{SE}\left(\theta \right)$, respectively.

The odd excitation vector can be determined as follows:(23)$$C_{{\left(N\times 1\right)}_{O}}={\left.C_{I\times 1}\right|}_{odd}$$
where the elements $C_{O}\left(n,1\right)$ of the vector $C_{{\left(N\times 1\right)}_{O}}$ can be obtained from the elements $C\left(i,1\right)$ of the vector $C_{I\times 1}$ such that:(24)$$C_{O}\left(n,1\right)=C\left(2i-1,1\right)$$
where $1\leq \left(i=n\right)\leq \frac{I+1}{2}$

The even excitation vector can be determined as follows:(25)$$C_{\left(N-1\right)\times 1_{E}}={\left.C_{I\times 1}\right|}_{even}$$
where the elements $C_{E}\left(n,1\right)$ of the vector $C_{\left(N-1\right)\times 1_{E}}$ can be obtained from the elements $C\left(i,1\right)$ of the vector $C_{I\times 1}$ such that:(26)$$C_{E}\left(n,1\right)=C\left(2i,1\right)$$
where $1\leq \left(i=n\right)\leq \left(\frac{I+1}{2}\right)-1$

Accordingly, the synthesized array factor using odd excitation coefficients ${AF}_{SO}\left(\theta \right)$ is given by:(27)$${AF}_{SO}\left(\theta \right)=\sum_{n=1}^{\frac{I+1}{2}}C_{O}\left(n,1\right)e^{\left(jk\left(n-1\right)d_{s}\cos\theta \right)}$$

While the synthesized array factor using even excitation coefficients ${AF}_{SE}\left(\theta \right)$ is given by:(28)$${AF}_{SE}\left(\theta \right)=\sum_{n=1}^{\left(\frac{I+1}{2}\right)-1}C_{E}\left(n,1\right)e^{\left(jk\left(n-1\right)d_{s}\cos\theta \right)}$$
where $d_{s}$ is the synthesized element spacing determined by the GA [13]. The GA optimizes the element spacing such that the synthesized array factors ${AF}_{SO}\left(\theta \right)$ and ${AF}_{SE}\left(\theta \right)$ provide the same HPBW as the original array factor $AF\left(\theta \right)$. It is worth mentioning that both synthesized ${AF}_{SO}\left(\theta \right)$ and ${AF}_{SE}\left(\theta \right)$ provide a twofold decrease in the SLL compared to the original array factor $AF\left(\theta \right)$.

After SLL reduction, the synthesized excitation coefficients $C_{{\left(N\times 1\right)}_{O}}$ and $C_{\left(N-1\right)\times 1_{E}}$ are kept constant and the EPP is performed for perturbing the elements’ positions in the azimuth plane to steer the main beam to the desired direction.

Accordingly, the synthesized array factor using odd excitation coefficients ${AF}_{SO}\left(\theta \right)$ and EPP can be expressed as:(29)$${AF}_{SO-EPP}\left(\theta \right)=\sum_{n=1}^{\frac{I+1}{2}}C_{O}\left(n,1\right)e^{\left(jk\left(n-1\right)d_{s}\cos\theta +jkd_{y_{n}}\sin\theta \right)}$$

While the synthesized array factor using even excitation coefficients ${AF}_{SE}\left(\theta \right)$ and EPP can be expressed as:(30)$${AF}_{SE-EPP}\left(\theta \right)=\sum_{n=1}^{\left(\frac{I+1}{2}\right)-1}C_{E}\left(n,1\right)e^{\left(jk\left(n-1\right)d_{s}\cos\theta +jkd_{y_{n}}\sin\theta \right)}$$
where $d_{y_{n}}$ is related to the synthesized element spacing $d_{s}$ according to Figure 26 shown below.

#### 2.2.1. Simulation Results of the Proposed SLL/EPP-BSA

Consider a broadside ULAA consisting of N=8 elements with uniform element spacing d=λ/2 whose elements are arranged along the *Z*-axis. Firstly, the ULAA is synthesized using the SC/GA for SLL reduction. The synthesized array factors ${AF}_{SO}\left(\theta \right)$ and ${AF}_{SE}\left(\theta \right)$ using odd and even excitations compared to the original array factor $AF\left(\theta \right)$ are shown in Figure 27 and Figure 28, respectively. The synthesized excitation coefficients $C_{{\left(N\times 1\right)}_{O}}$ and $C_{\left(N-1\right)\times 1_{E}}$, element spacing $d_{s}$, SLL, and HPBW are listed in Table 8.

#### 2.2.2. Simulation Results of the Proposed SLL/EPP-BSA for Five Test Case

**Test case (1):** R=1

In this case, the cosine wave half period is chosen to be equal to the array length, $H=L_{h}=4.935\;\lambda$, hence R=1. Accordingly, the EPPs of the synthesized array elements for $A_{c}$ changing from positive to negative are shown Figure 29. The relation between the amplitude changes of the cosine waveform and the scanning angle, HPBW, SLL, and maximum scanning range for the eight-element ULAA at a compression ratio R=1 are listed in Table 9. When the cosine wave amplitude is changing from positive to negative and its magnitude is changing from $A_{c}=0$ to $A_{c}=\lambda$, the array main beam tilted from a broadside direction of θ°=90° to θ°=62.36° with minimal changes in the HPBW that is increased by 0.12° at θ°=62.36° compared to the broadside direction, while the SLL increased from −29.79 dB to −10.53 dB as the main beam is titled away from the broadside direction to θ°=62.36°. The achieved maximum scanning range is $-27.64{}^{\circ}\leq \theta_{r}\leq 27.64{}^{\circ}$ around the broadside direction. The polar plot of the synthesized radiation pattern using the SLL/EPP-BSA approach at compression ratio R=1 and $A_{c}=\lambda$ is shown in Figure 30. This technique might be beneficial in mobile network applications when coverage is limited, and the coverage area is divided into sectors.

**Test case (2):** R=0.875

In this case, the cosine wave half period is chosen to be greater than the array length, $H>L_{h}$ and equals H=5.64 λ, hence R=4.935 λ/5.64 λ=0.875. Accordingly, after applying the EPPs of the synthesized array elements for $A_{c}$ changing from positive and the cosine wave amplitude is changing from positive to negative and its magnitude is changing from $A_{c}=0$ to $A_{c}=1.25\;\lambda$, the array main beam tilted from a broadside direction of θ°=90° to θ°=59.27° with minimal changes in the HPBW that decreased by 0.34° at θ°=59.27° compared to the broadside direction, while the SLL increased from −29.79 dB to −10.83 dB as the main beam titled away from the broadside direction to θ°=59.27°. The achieved maximum scanning range is $-30.73{}^{\circ}\leq \theta_{r}\leq 30.73{}^{\circ}$ around the broadside direction. The relation between the amplitude changes of the cosine waveform and the scanning angle, HPBW, SLL, and maximum scanning range for the eight-element ULAA at a compression ratio R=0.875 are listed in Table 10. The polar plot of the synthesized radiation pattern using the SLL/EPP-BSA approach at compression ratio R=0.875 and $A_{c}=1.25\;\lambda$ is shown in Figure 31.


**Test case (3): **

R=0.7778



In this case, the cosine wave half period is chosen to be greater than the array length, $H>L_{h}$ and equals H=6.345 λ, hence R=4.935λ/6.345λ=0.7778. Accordingly, after applying the EPPs of the synthesized array elements for $A_{c}$ changing from positive and the cosine wave amplitude is changing from positive to negative and its magnitude is changing from $A_{c}=0$ to $A_{c}=1.25\lambda$, the array main beam tilted from the broadside direction of θ°=90° to θ°=62.36° with minimal changes in the HPBW that is increased by 0.23° at θ°=62.36° compared to the broadside direction, while the SLL is increased from −29.79 dB to −19.1 dB as the main beam is titled away from the broadside direction to θ°=62.36°. The achieved maximum scanning range is $-27.64{}^{\circ}\leq \theta_{r}\leq 27.64{}^{\circ}$ around the broadside direction. The relation between the amplitude changes of the cosine waveform and the scanning angle, HPBW, SLL, and maximum scanning range for the eight-element ULAA at a compression ratio R=0.7778 are listed in Table 11. The polar plot of the synthesized radiation pattern using the SLL/EPP-BSA approach at compression ratio R=0.7778 and $A_{c}=1.25\;\lambda$ is shown in Figure 32.


**Test case (4): **

R=0.7



In this case, the cosine wave half period is chosen to be greater than the array length, $H>L_{h}$ and equals H=7.05 λ, and the array length equal $L_{h}=4.935\;\lambda$, hence R=0.7. Accordingly, after applying the EPPs of the synthesized array elements for $A_{c}$ changing from positive and the cosine wave amplitude is changing from positive to negative and its magnitude is changing from $A_{c}=0$ to $A_{c}=1.5\lambda$, the array main beam tilted from the broadside direction of θ°=90° to θ°=60.81° with minimal changes in the HPBW that is increased by 0.89° at $\theta {}^{\circ}=60.81^{}$ compared to the broadside direction, while SLL is increased from −29.79 dB to −12.68 dB as the main beam is titled away from the broadside direction to θ°=60.81°. The achieved maximum scanning range is $-29.19{}^{\circ}\leq \theta_{r}\leq 29.19{}^{\circ}$ around the broadside direction. The relation between the amplitude changes of the cosine waveform and the scanning angle, HPBW, SLL, and maximum scanning range for the eight-element ULAA at a compression ratio R=0.7 are listed in Table 12. The polar plot of the synthesized radiation pattern using the SLL/EPP-BSA approach at compression ratio R=0.7 and $A_{c}=1.5\;\lambda$ is shown in Figure 33.


**Test case (5): **

R=0.5



In this case, the cosine wave half period is chosen to be greater than the array length, H=9.87 λ, and the array length equal $L_{h}=4.935\;\lambda$, hence R=0.5. Accordingly, after applying the EPPs of the synthesized array elements for $A_{c}$ changing from positive and the cosine wave amplitude is changing from positive to negative and its magnitude is changing from $A_{c}=0$ to $A_{c}=2\;\lambda$, the array main beam tilted from the broadside direction of θ°=90° to θ°=65.97° with minimal changes in the HPBW that is increased by 3.02° at θ°=65.97° compared to the broadside direction, while the SLL is increased from −29.79 dB to −27.35 dB as the main beam is titled away from the broadside direction to θ°=65.97°. The achieved maximum scanning range is $-24.03{}^{\circ}\leq \theta_{r}\leq 24.03{}^{\circ}$ around the broadside direction. The relation between the amplitude changes of the cosine waveform and the scanning angle, HPBW, SLL, and maximum scanning range for the eight-element ULAA at a compression ratio R=0.5 are listed in Table 13. The polar plot of the synthesized radiation pattern using the SLL/EPP-BSA approach at compression ratio R=0.5 and $A_{c}=2\;\lambda$ is shown in Figure 34.

#### 2.2.3. Comparison between EPP/BSA and SLL/EPP-BSA

Based on prior simulations, we can conclude that the proposed EPP-BSA provides wider scanning ranges than the proposed SLL/EPP-BSA but with much higher SLLs at the endpoints of the scanning ranges. In this regard, the proposed SLL/EPP-BSA may be useful in interference-sensitive applications that require small scanning sectors.

## 3. Realization of EPP-BSA Using Planar Antenna Array Thinning

It is possible to design a PAA with a large number of antennas packed into a tiny space on a chip to create a system for millimeter-wave applications when the size of the antenna element is very small. Consider the test case of the M= eight-element LAA with uniform element spacing $d_{z}=0.5\;\lambda$ and size $L_{h}=3.5\;\lambda$, where the cosine wave half period is chosen to be greater than the array size, $H>L_{h}$ and equals H=4λ, and R=3.5 λ/4 λ=0.875. For cosine wave amplitude $A_{c}$ changing from 2 λ to −2 λ in the *Y*-direction, its span will be ${2A}_{c}=4\;\lambda$. The realization of the eight-element LAA sector beam scanning using the proposed EPP-BSA approach can be implemented by constructing a PAA of dimensions $\left(4\;\lambda \times 3.5\;\lambda \right)$ in the *Y*-direction and the *Z*-direction, respectively. Taking a uniform element spacing in the *Y*-direction to be $d_{y}=0.1\;\lambda$ and a uniform element spacing in the *Z*-direction to be $d_{z}=0.5\;\lambda$, a ${(M}_{y}\times M_{z})=$ $\left(41\times 8\right)$ planar antenna array (PAA) can be constructed as shown in Figure 35, where $M_{y\;}$ is the number of elements in the *Y*-direction that can be calculated as follows:(31)$$M_{y\;}=(2A_{c}/0.1\;\lambda )+1=(4\lambda /0.1\lambda )+1=41\;elements$$

$M_{z\;}$ is the number of elements in the *Z*-direction that can be calculated as follows:(32)$$M_{z\;}=M=8\;elements$$

Rather than using a rotating motor or progressive phase shift techniques for beam scanning, the proposed PAA configuration allows for precise array thinning to select the desired active “ON” eight elements and turn off the other elements to configure the designed EPP pattern to achieve a specific scan angle with the desired radiation properties. However, there may be some errors due to the mismatch between the positions of the elements of the actual antenna array and the positions of the selected active eight elements on the PAA for the desired EPP pattern. So that a position quantization is performed to select the nearest element on the PAA that matches the corresponding element on the actual antenna array in order to eliminate these inaccuracies as much as feasible. Table 14 shows a comparison between the actual element positions in the *Y*-direction compared to the corresponding quantized element positions in the *Y*-direction at fixed element spacing in the *Z*-direction.

The proposed PAA thinning technique is superior to utilizing progressive phase shifters to scan the main beam from the broadside to the desired direction since the proposed technique is based on modifying the EPP pattern rather than adjusting the phase shifter. This simplifies the production process while also lowering the cost of the feeding network. Furthermore, the proposed technique eliminates interference and coupling between the main beam and the back lobe at any scanning angle by maintaining a 180 spacing angle between them. However, when utilizing the progressive phase shift technique, the spacing angle between the main beam and the back lobe decreases as the scan angle increases, resulting in increased interference and coupling between them, which indeed increases the HPBW as the main beam approaches the end-fire direction.

The implementation of the proposed (41×8) PAA configuration using the CST microwave studio is shown in Figure 36. The array is realized using the dipole antenna illustrated in Figure 22 with resonance frequency $f_{o}=1\;GHz$. Table 15 shows a comparison between the CST simulation results and MATLAB simulation results of the proposed EPP/BSA approach using the proposed PAA thinning and actual EPP/BSA indicating the resultant scan angle, SLL, and HPBW at R=0.875.

Figure 37 shows the CST simulated 3D radiation patterns using the PAA thinning for implementation of the proposed EPP-BSA approach for the eight-element ULAA at R=0.875, $A_{c}=0$, $A_{c}=0.5\;\lambda$, $A_{c}=\lambda$, $A_{c}=2\;\lambda$, and changing from positive to negative. While the corresponding polar plots are shown in Figure 38.

## 4. Comparison with State-of-the-Art Works

In this section, the proposed beam steering techniques are compared with the state-of-the-art works. The comparison is performed in terms of the main concept of the utilized technique, beam steering range, maximum SLL, HPBW, and the significant remarks (drawbacks and advantages) as mentioned in Table 16.

In [19], the main beam steering is performed using a three-bit phase shifter. The steering network was formed from a uniform eight-branch power divider with each terminal feeding a microstrip line equipped with a switching mechanism that enabled three-bit periodic phase shifting for beam steering. The antenna array elements are fed by the network via coupled feeding. While the main drawbacks of the work are: (i) A complicated feeding network where the progressive phase shift of each antenna element is needed to be quantized and mapped from 0° to −360° in the eight different states of the individual 3-bit phase shifter. (ii) Each three-bit phase shifter requires eight PIN diodes, DC coupling capacitors, and a DC biasing network that complicates the design and increases the feeding network cost. (iii) The realized SLL of the array is higher than the ideal case due to the phase quantization errors that reaches 22.5° in this quantized three-bit phase shifter. (v) In contrast to the ideal situation, when phase quantization was not used, this phase inaccuracy is the reason for the main beam direction mismatch. (iv) The array gain changes with the change in the main beam direction.

In [20], a switched beam antenna array has been introduced. In order to produce four distinct main beam directions, the antenna’s phase shifter made use of a 4 × 4 planar Butler matrix with phase variations of ±135° and ±45° between its outputs. While the main drawbacks of the work are: (i) It did not offer continuous beam scanning; it just offered switched beams in limited and distinct directions. (ii) The beamwidth of the four main beams ranges from 38.5° to 55.4° in the H-plane and 26° to 31.8° in the E-plane, respectively. (iii) The proposed method suffers from significant HPBW broadening. (v) The feeding network design is complicated. In [21], the authors presented a switched beam antenna array. The main beam steering is performed using a 4 × 4 Butler matrix phase-shifter. While the main drawbacks of the work are: (i) It did not offer continuous beam scanning; it just offered switched beams in limited and distinct directions. (ii) The feeding network design is simple.

In [22], beam steering has been performed by utilizing a metasurface lens array that is fed by a phased array with fewer phase shifters. While the main drawbacks of the work are: (i) The design of the antenna array is complicated. (ii) It requires phase shifters. (iii) The integration of the lens array and the phased array is a cumbersome process and greatly affects the impedance matching of the array system. In [23], analog beam steering has been performed based on Huygens’ metasurface pixel technique without using external phase shifters to the array elements. The required phase shifts between antenna elements were generated by integrating conventional probe-fed microstrip patch antenna arrays with anisotropic Huygens metasurface. In [24], the grid array antenna structure has been used to introduce switched beam steering. By switching the excitation of the elements at various locations, the array main beam can be steered, while it suffers from limited accuracy.

## 5. Conclusions

In this paper, two sector beam scanning approaches based on the EPP technique, denoted as the EPP-BSA and the SLL/EPP-BSA, are introduced. Based on prior simulations, we can conclude that the proposed EPP-BSA provides wider scanning ranges than the proposed SLL/EPP-BSA but with much higher SLLs at the endpoints of the scanning range. In this regard, the proposed SLL/EPP-BSA may be useful in interference-sensitive applications that require small scanning sectors. The proposed techniques are based on modifying the EPP pattern rather than using phase shifters while maintaining the elements excitations. This simplifies the production process and lowers the cost of the feeding network. Furthermore, the proposed technique eliminates interference and coupling between the main beam and the back lobe at any scanning angle by maintaining a 180 spacing angle between them. Furthermore, a proposed PAA thinning approach is proposed to realize the proposed EPP-BSA approach. It allows for precise array thinning to select the desired active “ON” antenna elements and turn off the other elements to configure the designed EPP pattern to achieve a specific scan angle with the desired radiation properties. The results are realized using the computer simulation technology (CST) microwave studio software package, which is highly matched to the MATLAB simulations.

## Figures and Tables

**Figure 1 sensors-23-06557-f001:**
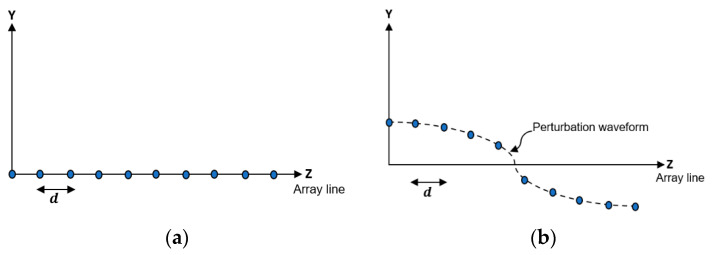
Linear antenna array (**a**) without EPP and (**b**) with EPP.

**Figure 2 sensors-23-06557-f002:**
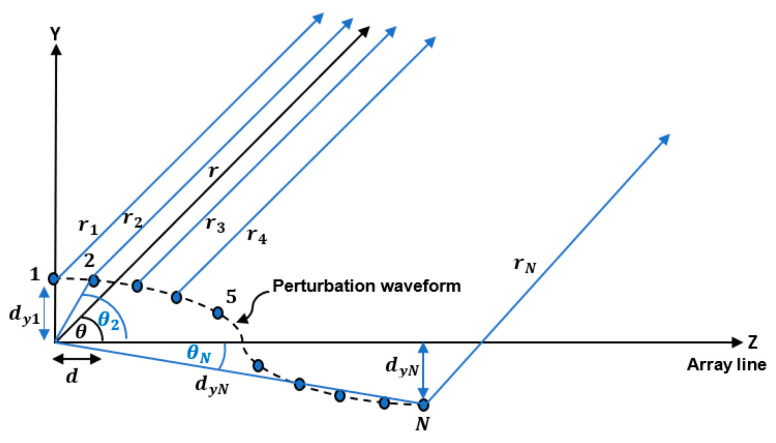
Geometry of a linear array of elements aligned along the *Z*-axis and has position perturbation along the *Y*-axis.

**Figure 3 sensors-23-06557-f003:**
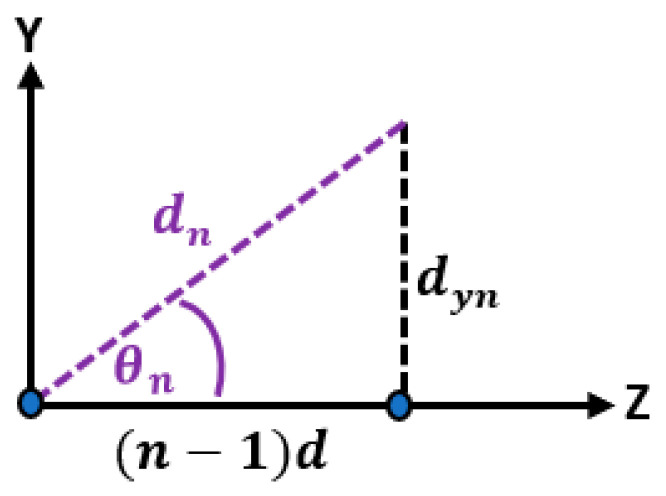
The triangular relation between the element position in the *Y*-direction and the distance from the reference antenna element.

**Figure 4 sensors-23-06557-f004:**
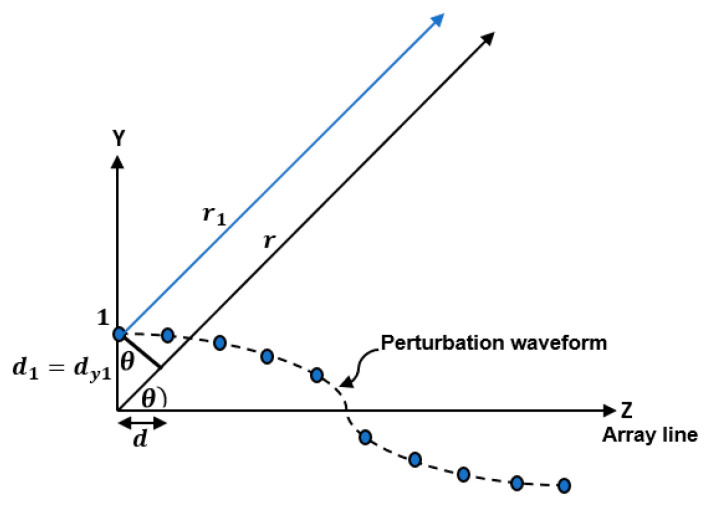
Geometry of a linear array of elements aligned along the *Z*-axis and has position perturbation along the *Y*-axis indicating the far field radiation from the first antenna element.

**Figure 5 sensors-23-06557-f005:**
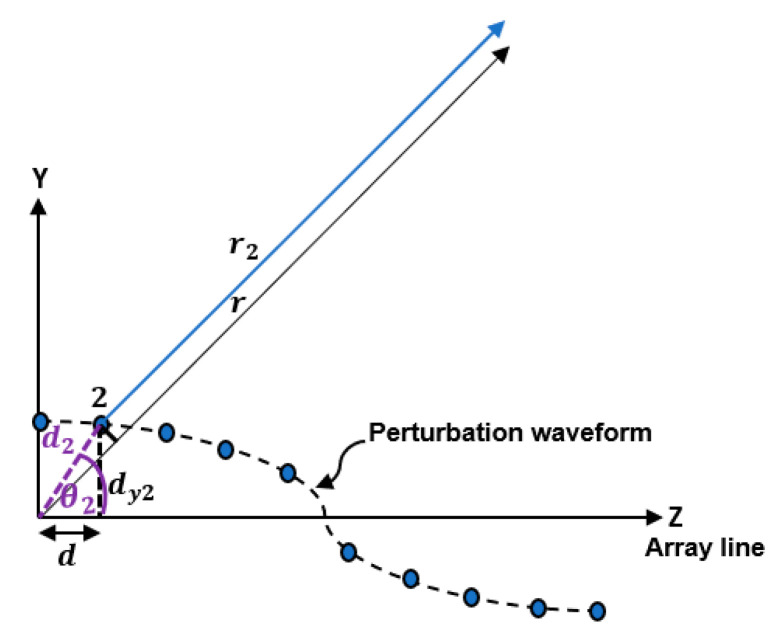
Geometry of a linear array of elements aligned along the *Z*-axis and has position perturbation along the *Y*-axis indicating the far field radiation from the second antenna element.

**Figure 6 sensors-23-06557-f006:**
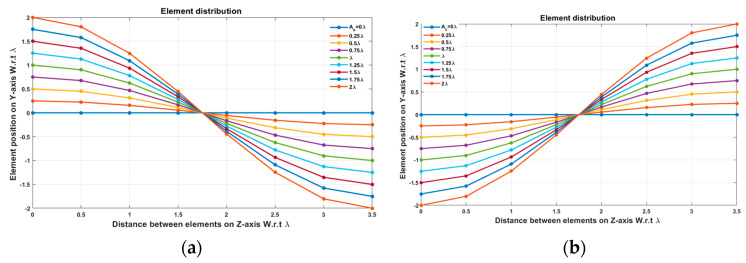
The EPPs of the array elements at compression ratio R=1 for (**a**) $A_{c}$ is changing from positive to negative and (**b**) $A_{c}$ is changing from negative to positive.

**Figure 7 sensors-23-06557-f007:**
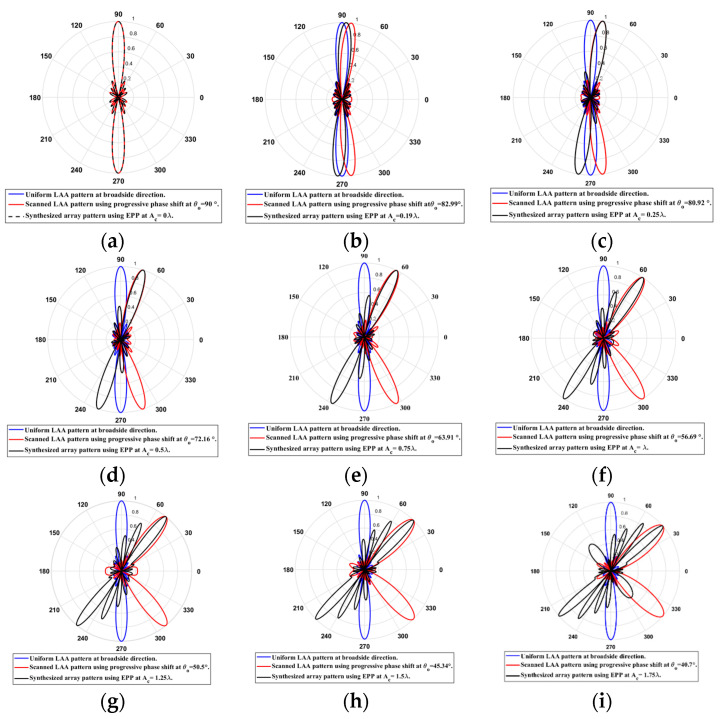
Polar plots of the synthesized radiation patterns using the EPP-BSA approach for 8-element LAA at compression ratio R=1 and (**a**) $A_{c}=0$, (**b**) $A_{c}=0.19\;\lambda$, (**c**) $A_{c}=0.25\;\lambda$, (**d**) $A_{c}=0.5\;\lambda$, (**e**) $A_{c}=0.75\;\lambda$, (**f**) $A_{c}=\lambda$, (**g**) $A_{c}=1.25\;\lambda$, (**h**)${\;A}_{c}=1.5\;\lambda$, (**i**)$\;A_{c}=1.75\;\lambda$, and (**j**) $A_{c}=2\;\lambda$ and changing from positive to negative.

**Figure 8 sensors-23-06557-f008:**
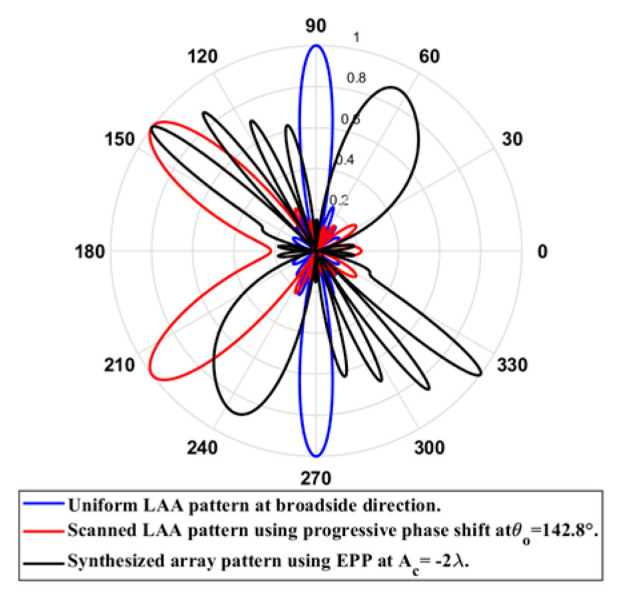
Polar plot of the synthesized radiation pattern using the EPP-BSA approach for 8-element ULAA at compression ratio R=1 for $A_{c}=2\;\lambda$ and changing from negative to positive.

**Figure 9 sensors-23-06557-f009:**
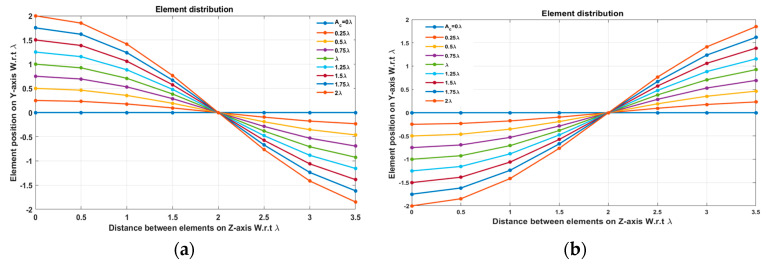
The EPPs of the array elements at compression ratio R=0.875 for (**a**) $A_{c}$ is changing from positive to negative and (**b**) $A_{c}$ is changing from negative to positive.

**Figure 10 sensors-23-06557-f010:**
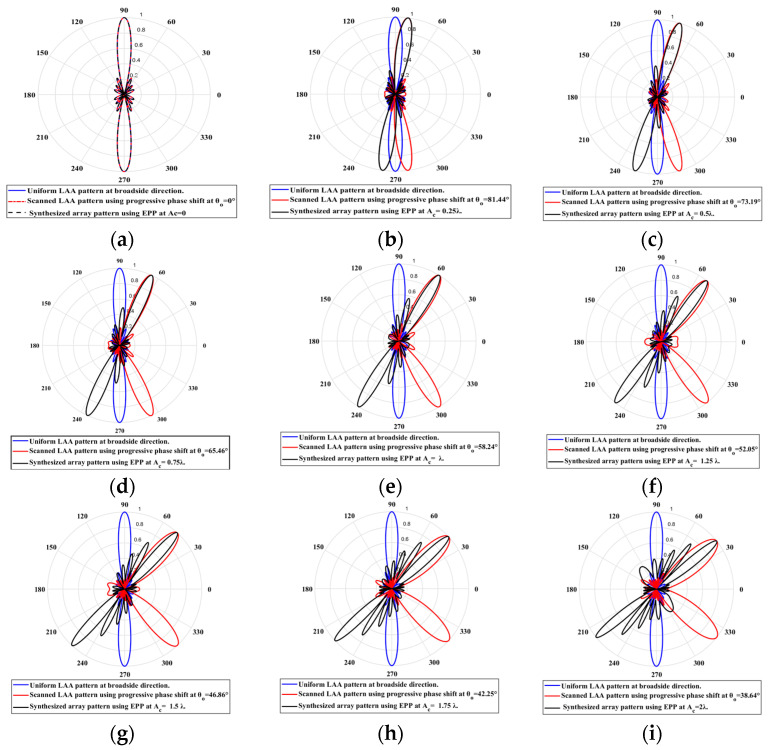
Polar plots of the synthesized radiation patterns using the EPP-BSA approach for an 8-element ULAA at compression ratio R=0.875 and (**a**) $A_{c}=0$, (**b**) $A_{c}=0.25\;\lambda$, (**c**) $A_{c}=0.5\;\lambda$, (**d**) $A_{c}=0.75\;\lambda$, (**e**) $A_{c}=\lambda$, (**f**) $A_{c}=1.25\;\lambda$, (**g**)${\;A}_{c}=1.5\;\lambda$, (**h**)$\;A_{c}=1.75\;\lambda$, and (**i**) $A_{c}=2\;\lambda$ and changing from positive to negative.

**Figure 11 sensors-23-06557-f011:**
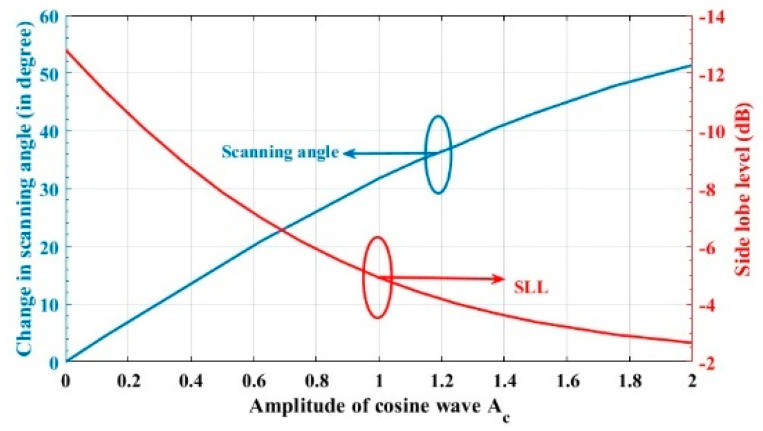
Relation between changes in the main beam scanning angle around the broadside direction and side lobe level changes.

**Figure 12 sensors-23-06557-f012:**
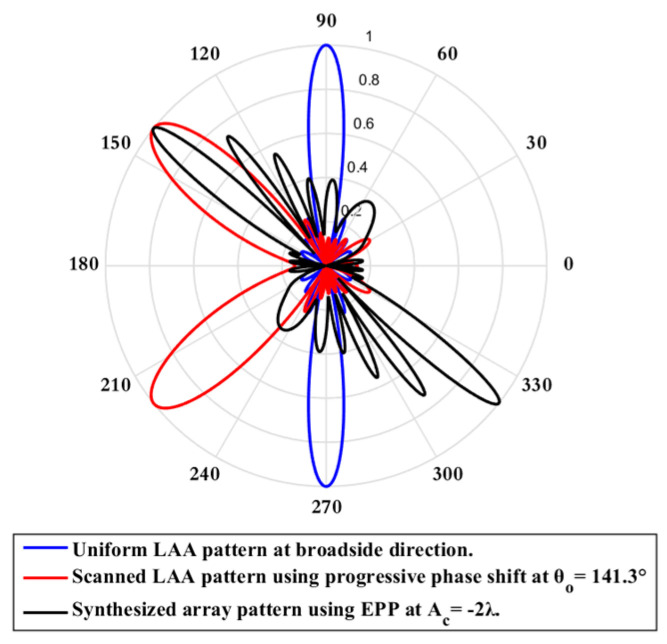
Polar plot of the synthesized radiation pattern using the EPP-BSA approach for 8-element ULAA at compression ratio R=0.875 for $A_{c}=2\;\lambda$ and changing from negative to positive.

**Figure 13 sensors-23-06557-f013:**
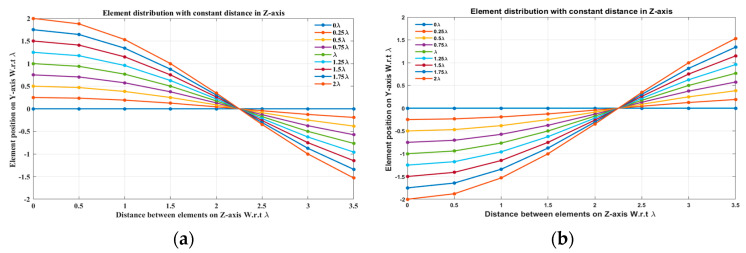
The EPPs of the array elements at compression ratio R=0.7778 for (**a**) $A_{c}$ is changing from positive to negative and (**b**) $A_{c}$ is changing from negative to positive.

**Figure 14 sensors-23-06557-f014:**
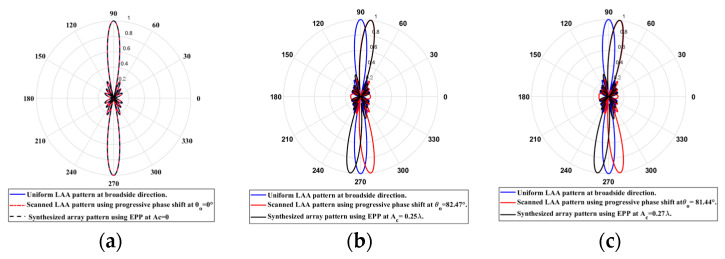
Polar plots of the synthesized radiation patterns using the EPP-BSA approach for an 8-element ULAA at compression ratio R=0.7778 at (**a**)$\;A_{c}=0$, (**b**) $A_{c}=0.25\;\lambda$, (**c**) $A_{c}=0.27\;\lambda$, (**d**) $A_{c}=0.3\;\lambda$ (**e**) $A_{c}=0.5\;\lambda$, (**f**) $A_{c}=0.75\;\lambda$, (**g**) $A_{c}=\lambda$, (**h**) $A_{c}=1.25\;\lambda$, (**i**)${\;A}_{c}=1.5\;\lambda$, (**j**)$\;A_{c}=1.75\;\lambda$, and (**k**) $A_{c}=2\;\lambda$ and changing from positive to negative.

**Figure 15 sensors-23-06557-f015:**
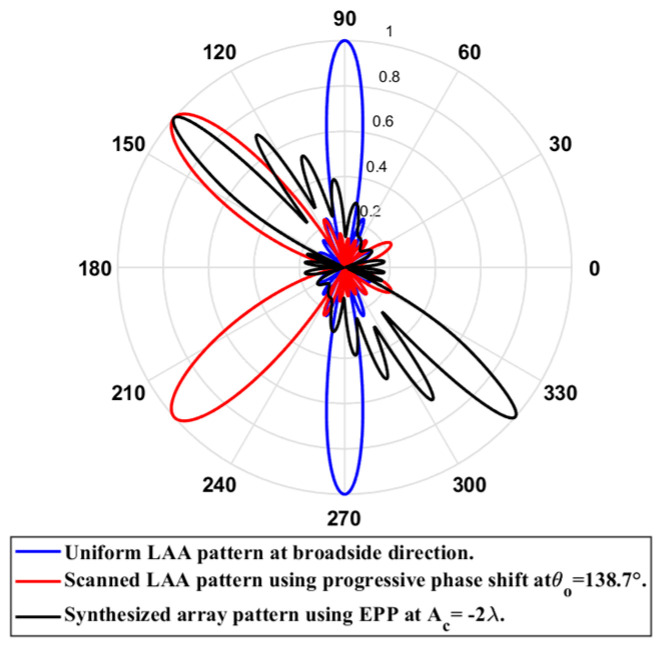
Polar plot of the synthesized radiation pattern using the EPP-BSA approach for 8-element ULAA at compression ratio R=0.7778 for $A_{c}=2\;\lambda$ and changing from negative to positive.

**Figure 16 sensors-23-06557-f016:**
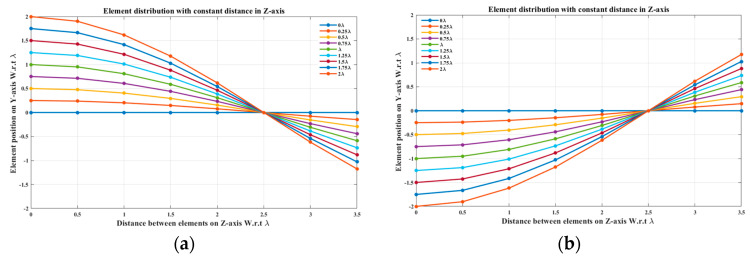
The EPPs of the array elements at compression ratio R=0.7 for (**a**) $A_{c}$ is changing from positive to negative and (**b**) $A_{c}$ is changing from negative to positive.

**Figure 17 sensors-23-06557-f017:**
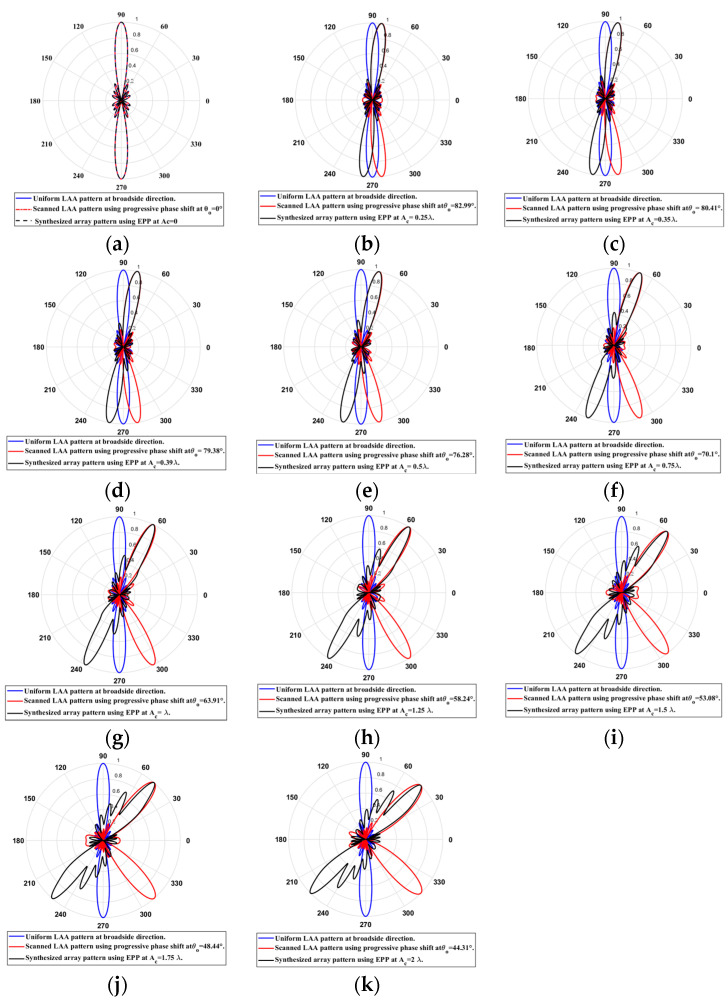
Polar plots of the synthesized radiation patterns using the EPP techniques for 8-element ULAA at compression ratio R=0.7 at: (**a**) $A_{c}=0$, (**b**) $A_{c}=0.25\;\lambda$, (**c**) $A_{c}=0.35\;\lambda$, (**d**) $A_{c}=0.39\;\lambda$, (**e**) $A_{c}=0.5\;\lambda$, (**f**) $A_{c}=0.75\;\lambda$, (**g**) $A_{c}=\lambda$, (**h**) $A_{c}=1.25\;\lambda$, (**i**)${\;A}_{c}=1.5\;\lambda$, (**j**)$\;A_{c}=1.75\;\lambda$, and (**k**) $A_{c}=2\;\lambda$ and changing from positive to negative.

**Figure 18 sensors-23-06557-f018:**
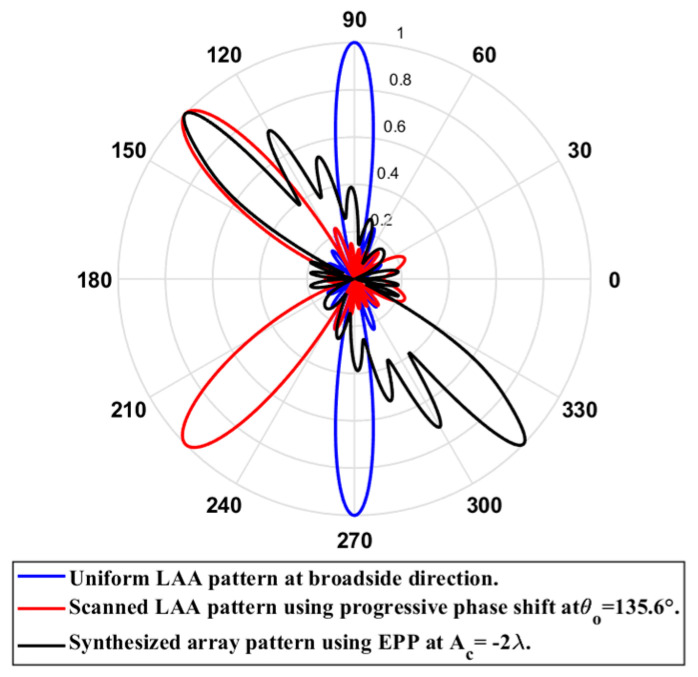
Polar plot of the synthesized radiation pattern using the EPP-BSA approach for 8-element ULAA at compression ratio R=0.7 for $A_{c}=2\;\lambda$ and changing from negative to positive.

**Figure 19 sensors-23-06557-f019:**
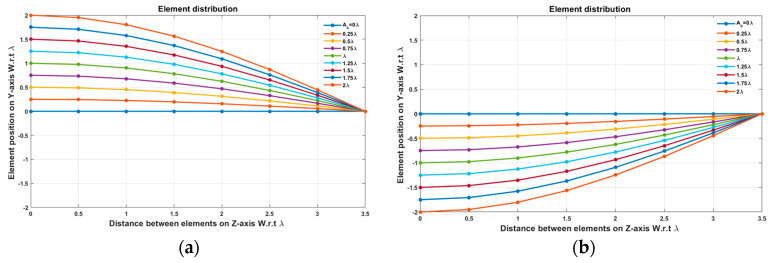
The EPPs of the array elements at compression ratio R=0.5 for (**a**) $A_{c}$ is changing from positive to negative and (**b**) $A_{c}$ is changing from negative to positive.

**Figure 20 sensors-23-06557-f020:**
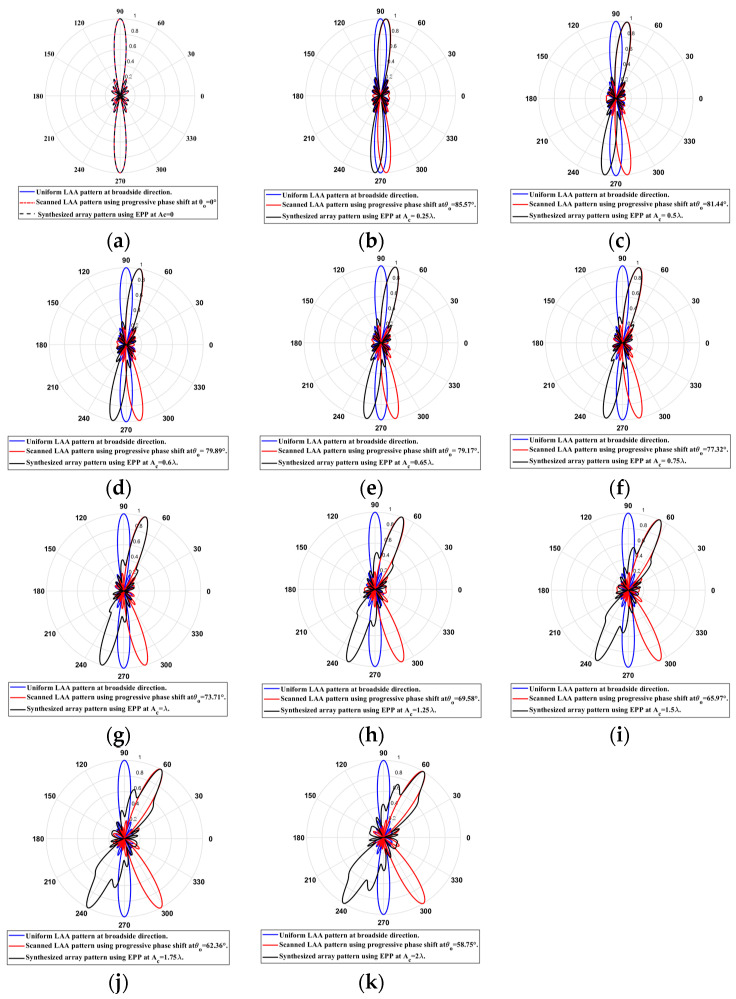
Polar plots of the synthesized radiation patterns using the EPP-BSA approach for 8-element uniform LAA at compression ratio equal to 0.5 at: (**a**) $A_{c}=0$, (**b**) $A_{c}=0.25\;\lambda$, (**c**) $A_{c}=0.5\;\lambda$, (**d**) $A_{c}=0.6\;\lambda$, (**e**) $A_{c}=0.65\;\lambda$, (**f**) $A_{c}=0.75\;\lambda$, (**g**) $A_{c}=\lambda$, (**h**) $A_{c}=1.25\;\lambda$, (**i**)${\;A}_{c}=1.5\;\lambda$, (**j**)$\;A_{c}=1.75\;\lambda$, and (**k**) $A_{c}=2\;\lambda$ and changing from positive to negative.

**Figure 21 sensors-23-06557-f021:**
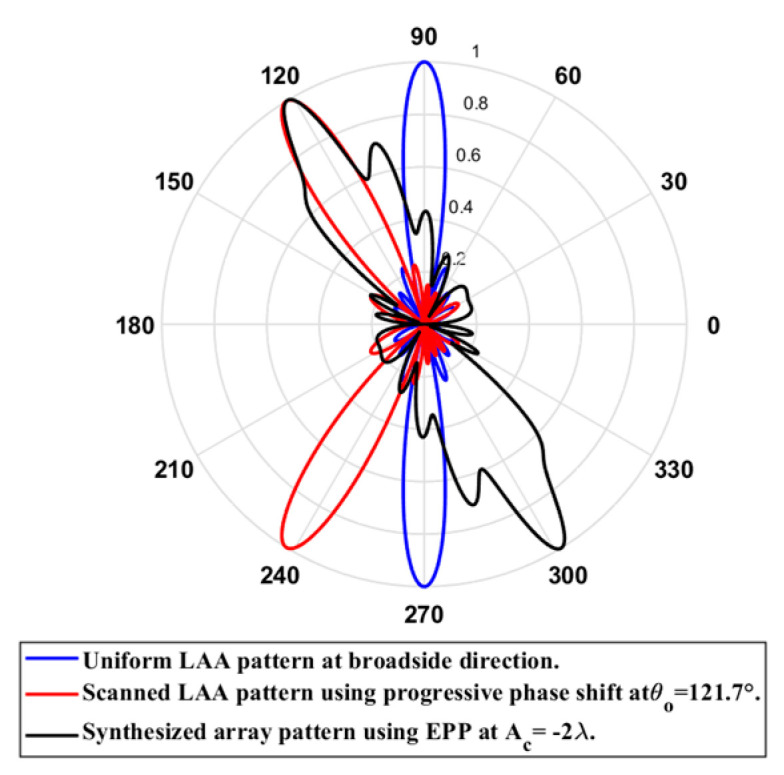
Polar plot of the synthesized radiation pattern using the EPP-BSA approach for 8-element uniform LAA due to compression ratio equal to 0.5 for $A_{c}=2\;\lambda$ and changing from negative to positive.

**Figure 22 sensors-23-06557-f022:**
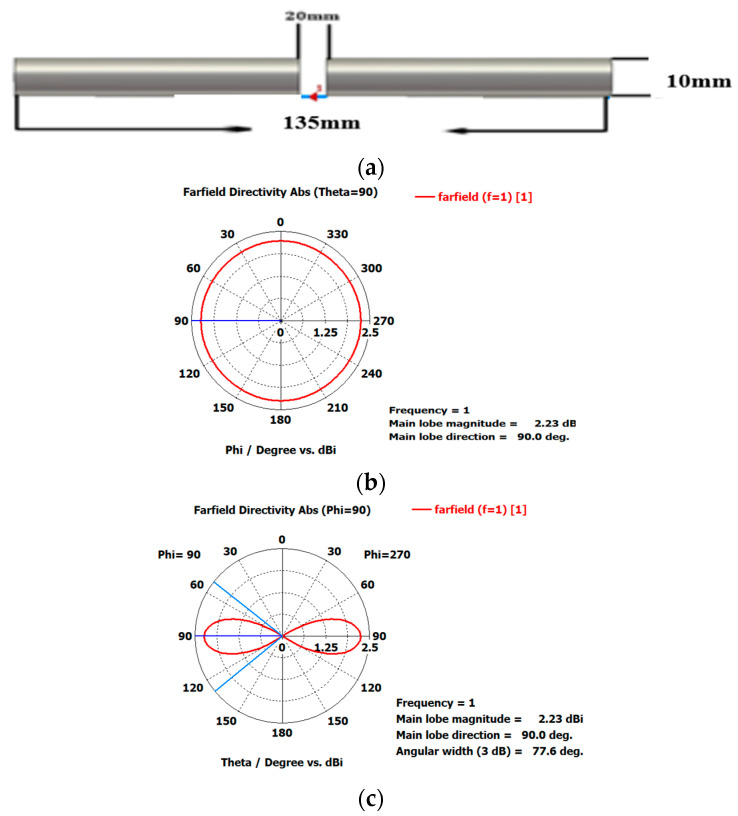
(**a**) Dimensions of dipole antenna, (**b**) H-plane pattern, and (**c**) E-plane pattern.

**Figure 23 sensors-23-06557-f023:**
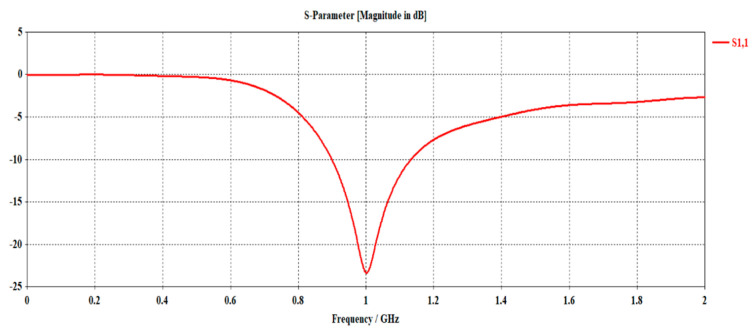
The scattering parameter (reflection coefficient) $\left|S_{11}\right|$ against the frequency for the designed dipole antenna.

**Figure 24 sensors-23-06557-f024:**
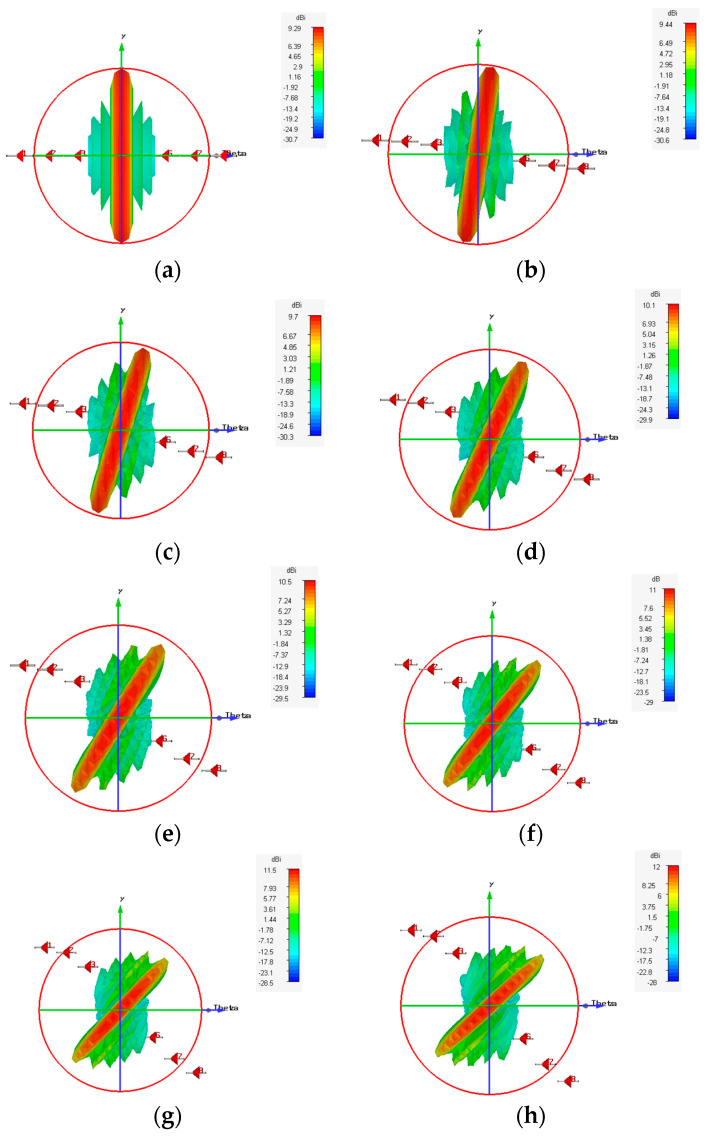
The 3D radiation pattern plots of the synthesized antenna arrays using CST software package applying the proposed EPP-BSA approach for an 8-element ULAA for R=0.875 at: (**a**) $A_{c}=0$, (**b**) $A_{c}=0.25\;\lambda$, (**c**) $A_{c}=0.5\;\lambda$, (**d**) $A_{c}=0.75\;\lambda$, (**e**) $A_{c}=1\;\lambda$, (**f**) $A_{c}=1.251\;\lambda$, (**g**) $A_{c}=1.5\;\lambda$, (**h**) $A_{c}=1.75\;\lambda$, (**i**) $A_{c}=2\;\lambda$, and changing from positive to negative.

**Figure 25 sensors-23-06557-f025:**
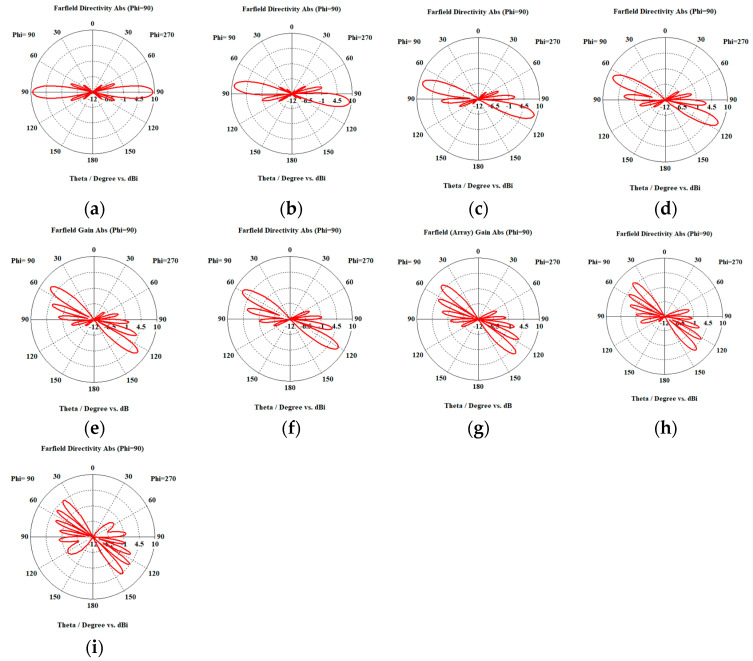
Polar plots of the synthesized antenna arrays radiation patterns using CST software package applying the proposed EPP-BSA approach for an 8-element ULAA for R=0.875 at: (**a**) $A_{c}=0$, (**b**) $A_{c}=0.25\lambda$, (**c**) $A_{c}=0.5\;\lambda$, (**d**) $A_{c}=0.75\;\lambda$, (**e**) $A_{c}=1\;\lambda$, (**f**) $A_{c}=1.251\;\lambda$, (**g**) $A_{c}=1.5\;\lambda$, (**h**) $A_{c}=1.75\;\lambda$, (**i**) $A_{c}=2\;\lambda$, and changing from positive to negative.

**Figure 26 sensors-23-06557-f026:**
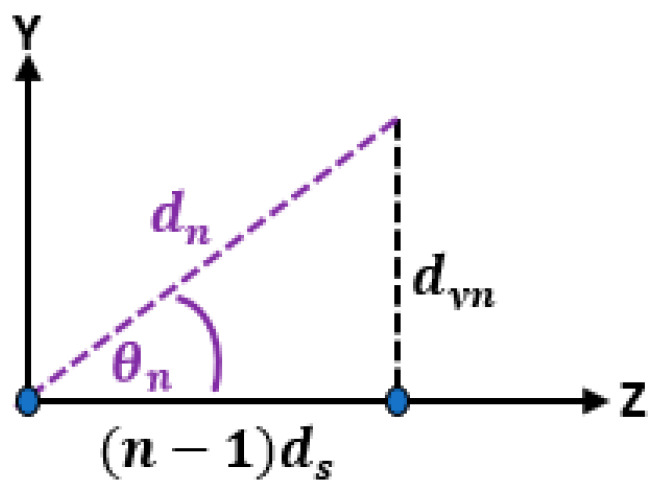
The triangular relation between the element position $d_{y_{n}}$ in the *Y*-direction and the distance from the reference antenna element.

**Figure 27 sensors-23-06557-f027:**
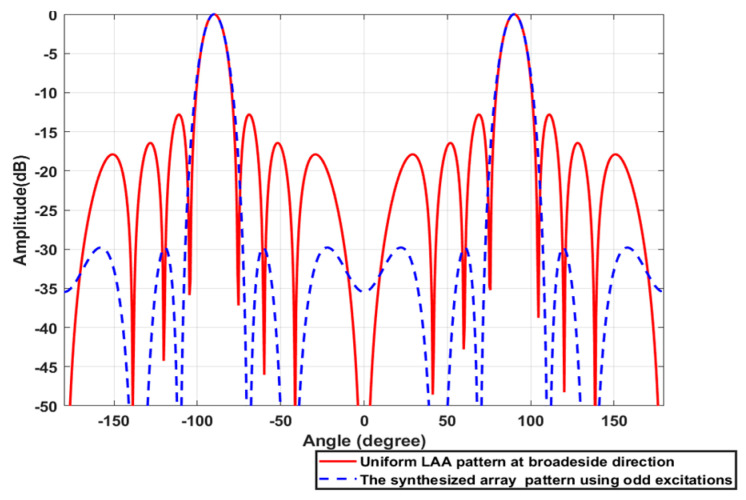
The synthesized array factor ${AF}_{SO}\left(\theta \right)$ using odd excitations compared to the original array factor $AF\left(\theta \right)$ of the 8-element ULAA.

**Figure 28 sensors-23-06557-f028:**
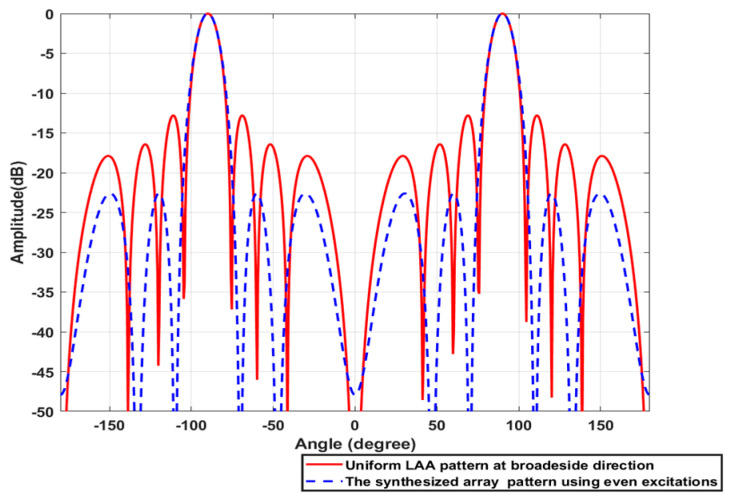
The synthesized array factor ${AF}_{SE}\left(\theta \right)$ using even excitations compared to the original array factor $AF\left(\theta \right)$ of the 8-element ULAA.

**Figure 29 sensors-23-06557-f029:**
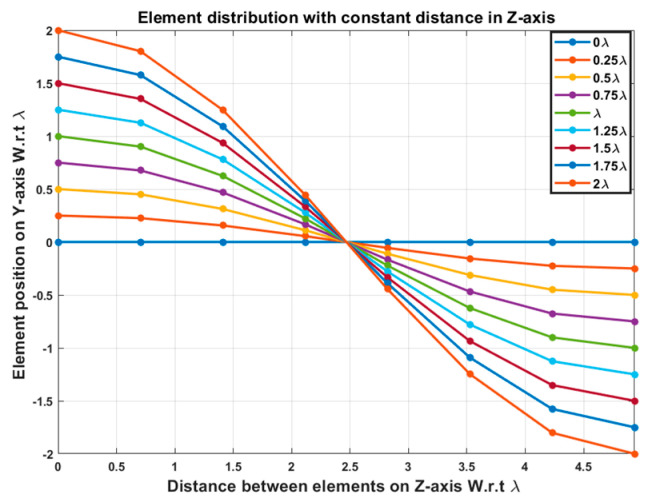
The EPPs of the synthesized array elements at compression ratio R=1 for $A_{c}$ is changing from positive to negative.

**Figure 30 sensors-23-06557-f030:**
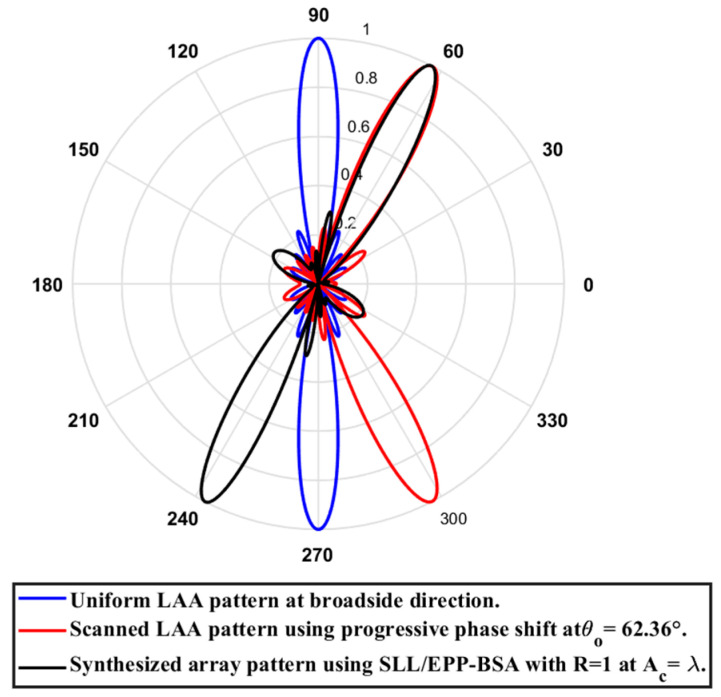
Polar plot of the synthesized radiation pattern using the SLL/EPP-BSA approach at compression ratio R=1 and $A_{c}=\lambda$.

**Figure 31 sensors-23-06557-f031:**
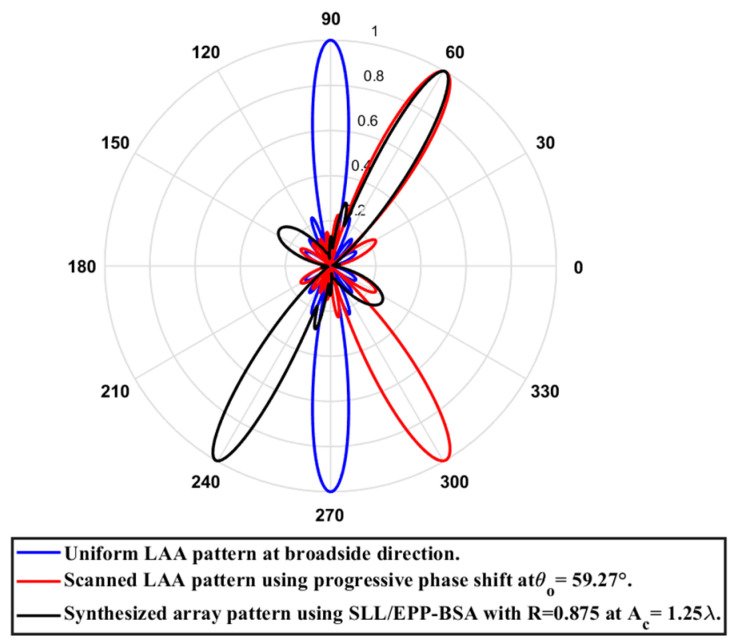
Polar plot of the synthesized radiation pattern using the SLL/EPP-BSA approach at compression ratio R=0.875 and $A_{c}=1.25\;\lambda$.

**Figure 32 sensors-23-06557-f032:**
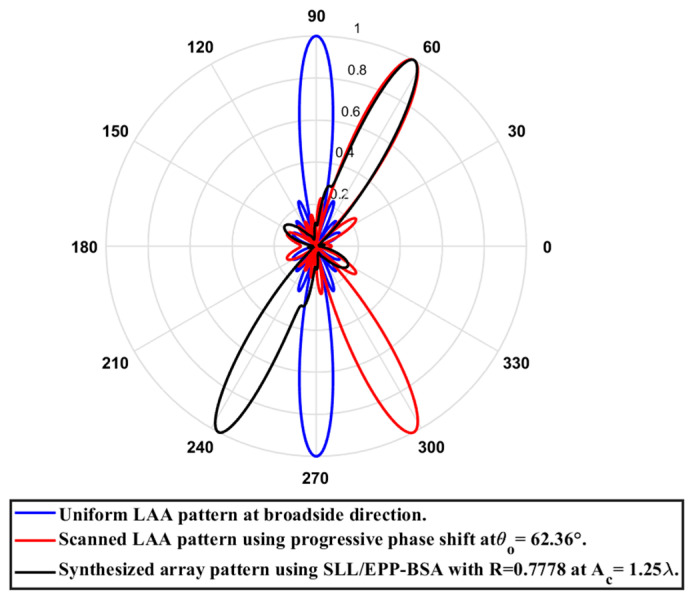
Polar plot of the synthesized radiation pattern using the SLL/EPP-BSA approach at compression ratio R=0.7778 and $A_{c}=1.25\;\lambda$.

**Figure 33 sensors-23-06557-f033:**
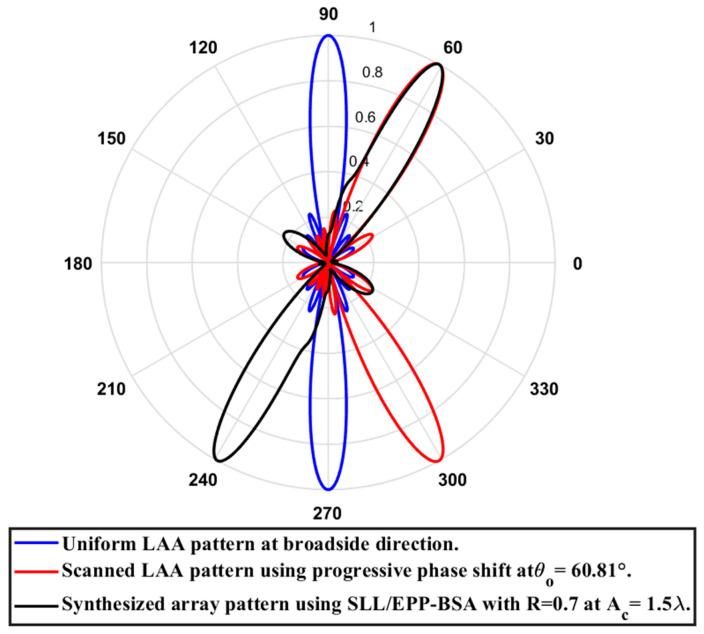
Polar plot of the synthesized radiation pattern using the SLL/EPP-BSA approach at compression ratio R=0.7 and $A_{c}=1.5\;\lambda$.

**Figure 34 sensors-23-06557-f034:**
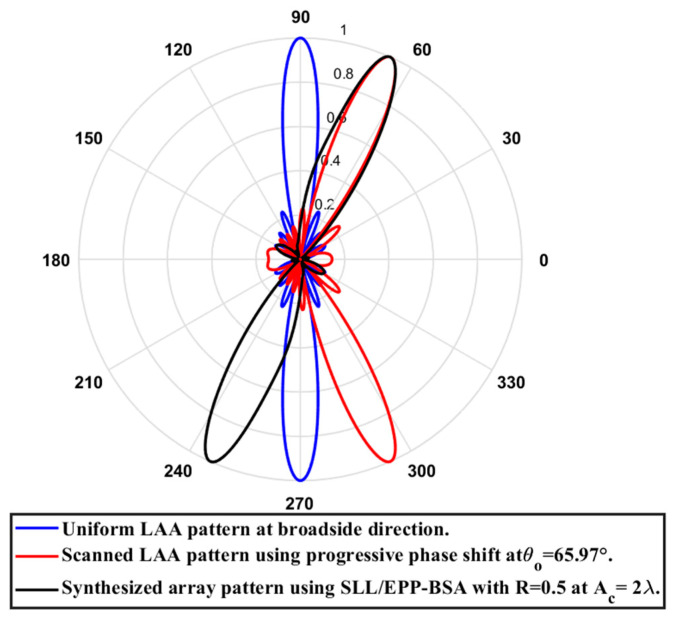
Polar plot of the synthesized radiation pattern using the SLL/EPP-BSA approach at compression ratio R=0.5 and $A_{c}=2\;\lambda$.

**Figure 35 sensors-23-06557-f035:**
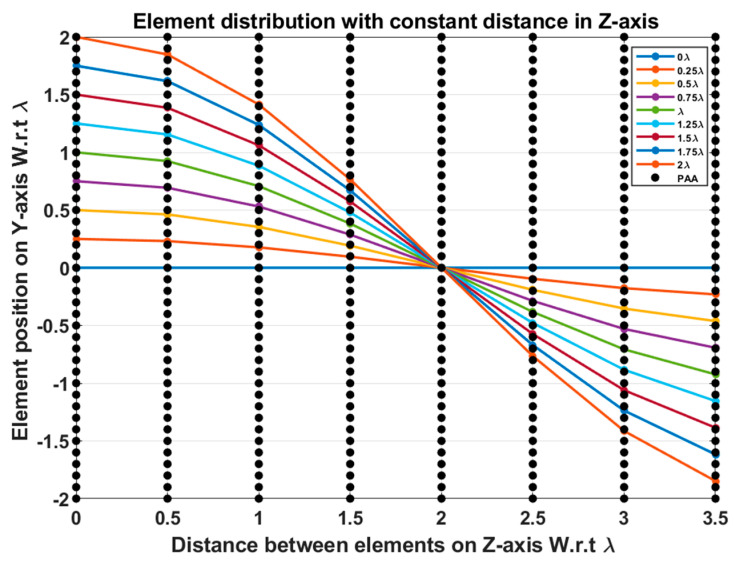
Element distribution of the proposed (41×8) PAA configuration and the element distributions of the M=8 elements of the actual array using the proposed EPP-BSA approach at uniform element spacing $d_{z}=0.5\;\lambda$ on the *Z*-axis and $A_{c}$ changing from positive to negative.

**Figure 36 sensors-23-06557-f036:**
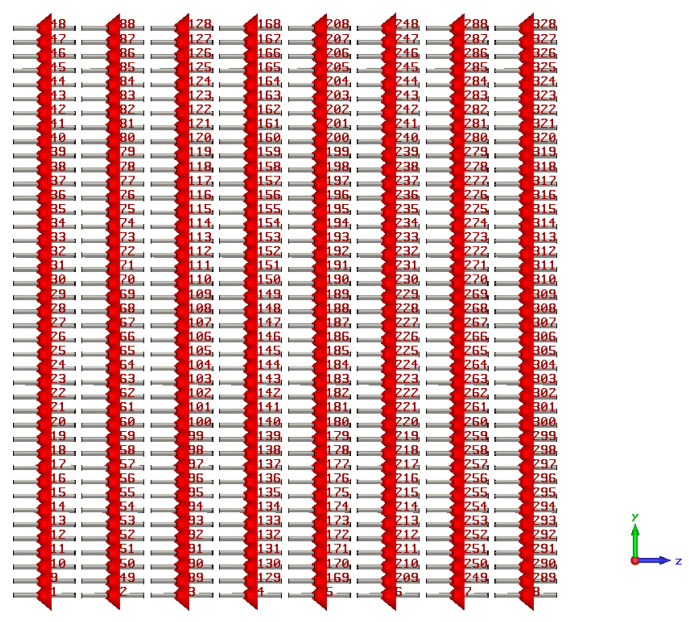
The implementation of the proposed (41×8) PAA configuration using CST microwave studio utilizing a dipole antenna with resonance frequency $f_{o}=1\;GHz$.

**Figure 37 sensors-23-06557-f037:**
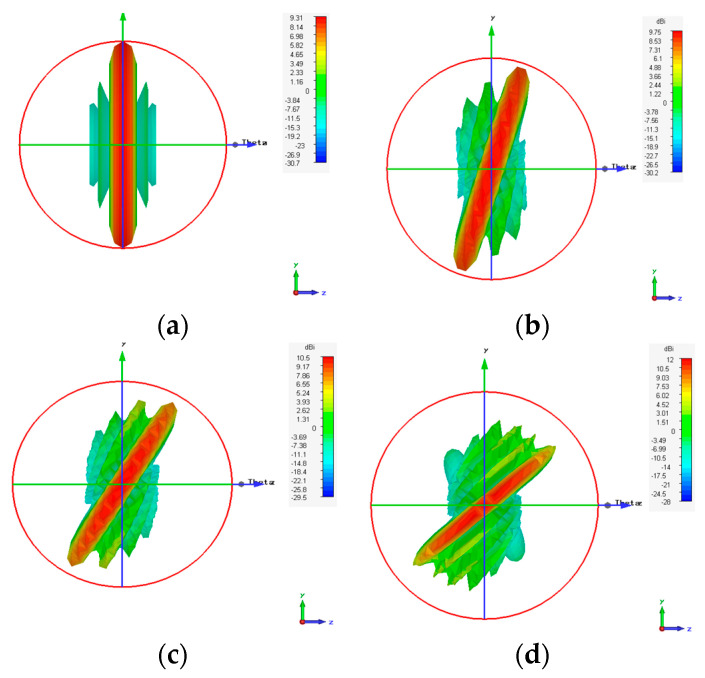
CST-simulated 3D radiation patterns using the PAA thinning for implementation of the proposed EPP-BSA approach for 8-element ULAA at R=0.875: (**a**) $A_{c}=0$, (**b**) $A_{c}=0.5\;\lambda$, (**c**) $A_{c}=\lambda$, (**d**) $A_{c}=2\;\lambda$, and changing from positive to negative.

**Figure 38 sensors-23-06557-f038:**
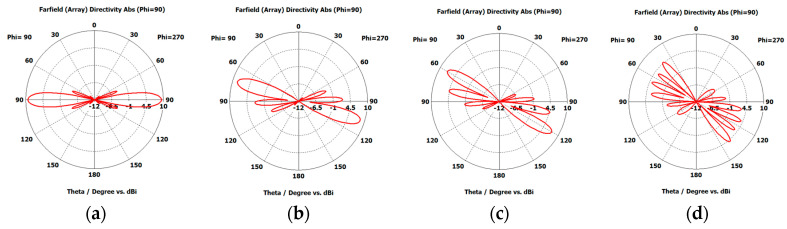
CST simulated polar plots of the radiation patterns using the PAA thinning for implementation of the proposed EPP-BSA approach for 8-element ULAA at R=0.875: (**a**) $A_{c}=0$, (**b**) $A_{c}=0.5\;\lambda$, (**c**) $A_{c}=\lambda$, (**d**) $A_{c}=2\;\lambda$, and changing from positive to negative.

**Table 1 sensors-23-06557-t001:** The relation between the amplitude changes of the cosine waveform and the scanning angle, HPBW, SLL, and maximum scanning range $\theta_{r}$ for 8-element uniform LAA at compression ratio R=1.

Figure	$\mathrm{Cosine}\;\mathrm{Wave}\;\mathrm{Amplitude}\;A_{c}$	$\mathrm{Main}\;\mathrm{Beam}\;\mathrm{Direction}\;\theta_{o}$, in °	SLL (dB)	HPBW in °	Maximum Scanning Range
Figure 7a	0	90°	−12.8 dB	12.8°	$-52.91{}^{\circ}\leq \theta_{r}\leq 52.91{}^{\circ}$
Figure 7b	0.19 λ	82.99°	−10.09 dB	12.67°	
Figure 7c	0.25 λ	80.92°	−9.346 dB	12.65°	
Figure 7d	0.5 λ	72.16°	−6.82 dB	12.33°	
Figure 7e	0.75 λ	63.91°	−4.963 dB	11.85°	
Figure 7f	λ	56.69°	−3.618 dB	11.1°	
Figure 7g	1.25 λ	50.5°	−2.671 dB	10.45°	
Figure 7h	1.5 λ	45.34°	−2.002 dB	9.8°	
Figure 7i	1.75 λ	40.7°	−1.454 dB	9.3°	
Figure 7j	2 λ	37.09°	−1.233 dB	8.75°	

**Table 2 sensors-23-06557-t002:** The relation between the amplitude changes of the cosine waveform and the scanning angle, HPBW, SLL, and maximum scanning range $\theta_{r}$ for 8-element ULAA at compression ratio R=0.875.

Figure	$\mathrm{Cosine}\;\mathrm{Wave}\;\mathrm{Amplitude}\;A_{c}$	$\mathrm{Main}\;\mathrm{Beam}\;\mathrm{Direction}\;\theta_{o}$, in °	SLL (dB)	HPBW in °	Maximum Scanning Range
Figure 10a	0	90°	−12.8 dB	12.8°	$-51.36{}^{\circ}\leq \theta_{r}\leq 51.36{}^{\circ}$
Figure 10b	0.25 λ	81.44°	−10.09 dB	12.6°	
Figure 10c	0.5 λ	73.19°	−7.88 dB	12.4°	
Figure 10d	0.75 λ	65.46°	−6.181 dB	12°	
Figure 10e	λ	58.24°	−4.917 dB	11.2°	
Figure 10f	1.25 λ	52.05°	−4.009 dB	10.8°	
Figure 10g	1.5 λ	46.86°	−3.386 dB	10.5°	
Figure 10h	1.75 λ	42.25°	−2.949 dB	9.7°	
Figure 10i	2 λ	38.64°	−2.668 dB	9.2°	

**Table 3 sensors-23-06557-t003:** The relation between the amplitude changes of the cosine waveform and the scanning angle, HPBW, SLL, and maximum scanning range $\theta_{r}$ for 8-element ULAA at compression ratio R=0.7778.

Figure	$\mathrm{Cosine}\;\mathrm{Wave}\;\mathrm{Amplitude}\;A_{c}$	$\mathrm{Main}\;\mathrm{Beam}\;\mathrm{Direction}\;\theta_{o}$, in °	SLL (dB)	HPBW in °	Maximum Scanning Range
Figure 14a	0	90°	−12.8 dB	12.8°	$-48.78{}^{\circ}\leq \theta_{r}\leq 48.78{}^{\circ}$
Figure 14b	0.25 λ	82.47°	−10.75 dB	12.7°	
Figure 14c	0.27 λ	81.44°	−10.55 dB	12.71°	
Figure 14d	0.3 λ	80.92°	−10.03 dB	12.68°	
Figure 14e	0.5 λ	74.74°	−8.679 dB	12.52°	
Figure 14f	0.75 λ	68.03°	−6.93 dB	12.21°	
Figure 14g	λ	61.33°	−5.567 dB	11.96°	
Figure 14h	1.25 λ	55.14°	−4.547 dB	11.77°	
Figure 14i	1.5 λ	49.99°	−3.842 dB	11.61°	
Figure 14j	1.75 λ	45.34°	−3.366 dB	11.5°	
Figure 14k	2 λ	41.22°	−3.097 dB	11.42°	

**Table 4 sensors-23-06557-t004:** The relation between the amplitude changes of the cosine waveform and the scanning angle, HPBW, SLL, and maximum scanning range $\theta_{r}$ for 8-element ULAA at compression ratio R=0.7.

Figure	$\mathrm{Cosine}\;\mathrm{Wave}\;\mathrm{Amplitude}\;A_{c}$	$\mathrm{Main}\;\mathrm{Beam}\;\mathrm{Direction}\;\theta_{o}$, in °	SLL (dB)	HPBW in °	Maximum Scanning Range
Figure 17a	0	90°	−12.8 dB	12.8°	$-45.69{}^{\circ}\leq \theta_{r}\leq 45.69{}^{\circ}$
Figure 17b	0.25 λ	82.99°	−11.2 dB	12.74°	
Figure 17c	0.35 λ	80.41°	−10.36 dB	12.66°	
Figure 17d	0.39 λ	79.38°	−10.05 dB	12.66°	
Figure 17e	0.5 λ	76.28°	−9.198 dB	12.65°	
Figure 17f	0.75 λ	70.1°	−7.39 dB	12.5°	
Figure 17g	λ	63.91°	−5.869 dB	12.42°	
Figure 17h	1.25 λ	58.24°	−4.68 dB	12.4°	
Figure 17i	1.5 λ	53.08°	−3.791 dB	12.7°	
Figure 17j	1.75 λ	48.44°	−3.185 dB	13.4°	
Figure 17k	2 λ	44.31°	−2.819 dB	13.9°	

**Table 5 sensors-23-06557-t005:** The relation between the amplitude changes of the cosine waveform and the scanning angle, HPBW, SLL, and maximum scanning range $\theta_{r}$ for 8-element ULAA at compression ratio R=0.5.

Figure	$\mathrm{Cosine}\;\mathrm{Wave}\;\mathrm{Amplitude}\;A_{c}$	$\mathrm{Main}\;\mathrm{Beam}\;\mathrm{Direction}\;\theta_{o}$, in °	SLL (dB)	HPBW in °	Maximum Scanning Range
Figure 19a	0	90°	−12.8 dB	12.8°	$-31.25{}^{\circ}\leq \theta_{r}\leq 31.25{}^{\circ}$
Figure 19b	0.25 λ	85.57°	−12.08 dB	12.79°	
Figure 19c	0.5 λ	81.44°	−10.89 dB	12.79°	
Figure 19d	0.6 λ	79.89°	−10.32 dB	12.77°	
Figure 19e	0.65 λ	79.17°	−10.02 dB	12.76°	
Figure 19f	0.75 λ	77.32°	−9.416 dB	12.8°	
Figure 19g	λ	73.71°	−7.874 dB	12.89°	
Figure 19h	1.25 λ	69.58°	−6.407 dB	13°	
Figure 19i	1.5 λ	65.97°	−5.087 dB	13.328°	
Figure 19j	1.75 λ	62.36°	−3.915 dB	13.94°	
Figure 19k	2 λ	58.75°	−2.929 dB	15.63°	

**Table 6 sensors-23-06557-t006:** Comparison between the five test cases of compression ratio. The optimal value is in bold.

No.	Compression Ratio	$\mathrm{Cosine}\;\mathrm{Wave}\;\mathrm{Amplitude}\;A_{c}$	Scan Angle Range with Respect to Broadside Direction	SLL	HPBW
1	R=1	0 to 0.19 λ	From 0° to 7.01°	<−10 dB	Constant
		0.19λ to 1.5 λ	From 7.01° to 52.91°	>−10 dB	Decreased by 3°
2	R=0.875	0 to 0.25 λ	From 0° to 8.56°	<−10 dB	Constant
		0.25 λ to 2 λ	From 8.5°to 51.36°	>−10 dB	Decreased by 3.6°
3	R=0.7778	0 to 0.3 λ	From 0°to 9.08°	<−10 dB	Constant
		0.3 λ to 2 λ	From 9.08° to 48.78°	>−10 dB	Decreased by 1.38°
4	R=0.7	0 to 0.39 λ	From 0° to 10.62°	<−10 dB	Constant
		0.39 λ to 2 λ	From 10.62° to 45.69°	>−10 dB	Increase by 1.1°
5	R=0.5	0 to 0.65 λ	From 0° to 10.83°	<−10 dB	Constant
		0.65 λ to 2 λ	From 10.62° to 31.25°	>−10 dB	Increased by 2.83°

**Table 7 sensors-23-06557-t007:** The CST simulation results for $A_{c}$ changing from 0 to 2 λ compared to the MATLAB simulation results indicating the resultant scan angle, SLL, and HPBW.

$\mathrm{Cosine}\;\mathrm{Wave}\;\mathrm{Amplitude}\;A_{c}$	CST Results			MATLAB Results		
	Scan Angle	HPBW	SLL	Scan Angle	HPBW	SLL
$A_{c}=0$	0°	12.3°	−13 dB	0°	12.8°	−12.8 dB
0.25 λ	8°	12.2°	−12.2 dB	8.56°	12.6°	−10.09 dB
0.5 λ	17°	12°	−7.5 dB	16.81°	12.4°	−7.88 dB
0.75 λ	24°	11.6°	−5.6 dB	24.54°	12°	−6.181 dB
λ	31°	11.2°	−4.1 dB	31.76°	11.2°	−4.917 dB
1.25 λ	37°	10.7°	−3.2 dB	37.95°	10.8°	−4.009 dB
1.5 λ	43°	10.2°	−2.1 dB	43.14°	10.5°	−3.386 dB
1.75 λ	47°	9.7°	−1.3 dB	47.75°	9.7°	−2.949 dB
2 λ	51°	9.1°	−2.1 dB	51.36°	9.2°	−2.668 dB

**Table 8 sensors-23-06557-t008:** The synthesized excitation coefficients $C_{{\left(N\times 1\right)}_{O}}$ and $C_{\left(N-1\right)\times 1_{E}}$, element spacing $d_{s}$, SLL, and HPBW.

Original ULAA		The Synthesized Array Using Odd Excitations		The Synthesized Array Using Even Excitations	
N = 8 elements		N = 8 elements		N = 7 elements	
$d_{s}\backslash \lambda =0.5$		$d_{s}\backslash \lambda =0.705$		$d_{s}\backslash \lambda =0.735$	
$a_{1}$	1	$a_{1}$	1	$a_{1}$	2
$a_{2}$	1	$a_{2}$	3	$a_{2}$	4
$a_{3}$	1	$a_{3}$	5	$a_{3}$	6
$a_{4}$	1	$a_{4}$	7	$a_{4}$	8
$a_{5}$	1	$a_{5}$	7	$a_{5}$	6
$a_{6}$	1	$a_{6}$	5	$a_{6}$	4
$a_{7}$	1	$a_{7}$	3	$a_{7}$	2
$a_{8}$	1	$a_{8}$	1		
HPBW=12.8°		HPBW=12.8°		HPBW=12.8°	
SLL=−12.8 dB		SLL=−29.79 dB		SLL=−22.61 dB	

**Table 9 sensors-23-06557-t009:** The relation between the amplitude changes of the cosine waveform and the scanning angle, HPBW, SLL, and maximum scanning range $\theta_{r}$ for 8-element ULAA at compression ratio R=1.

$\mathrm{Cosine}\;\mathrm{Wave}\;\mathrm{Amplitude}\;A_{c}$	$\mathrm{Main}\;\mathrm{Beam}\;\mathrm{Direction}\;\theta_{o}$, in °	SLL (dB)	HPBW in °	Maximum Scanning Range
0	90°	−29.79 dB	12.8°	$-27.64{}^{\circ}\leq \theta_{r}\leq 27.64{}^{\circ}$
0.25 λ	82.99°	−22.94 dB	12.78°	
0.5 λ	75.77°	−16.46 dB	12.83°	
0.75 λ	69.06°	−12.84 dB	12.89°	
λ	62.36°	−10.53 dB	12.92°	
1.25 λ	Grating lobe appearance			
1.5 λ				
1.75 λ				
2 λ				

**Table 10 sensors-23-06557-t010:** The relation between the amplitude changes of the cosine waveform and the scanning angle, HPBW, SLL, and maximum scanning range $\theta_{r}$ for 8-element ULAA at compression ratio R=0.875.

$\mathrm{Cosine}\;\mathrm{Wave}\;\mathrm{Amplitude}\;A_{c}$	$\mathrm{Main}\;\mathrm{Beam}\;\mathrm{Direction}\;\theta_{o}$, in °	SLL (dB)	HPBW in °	Maximum Scanning Range
0	90°	−29.79 dB	12.8°	$-30.73{}^{\circ}\leq \theta_{r}\leq 30.73{}^{\circ}$
0.25 λ	83.5°	−25.55 dB	12.76°	
0.5 λ	76.8°	−18.78 dB	12.77°	
0.75 λ	70.61°	−14.96 dB	12.75°	
λ	64.94°	−12.5 dB	12.67°	
1.25 λ	59.27°	−10.83 dB	12.46°	
1.5 λ	Grating lobe appearance			
1.75 λ				
2 λ				

**Table 11 sensors-23-06557-t011:** The relation between the amplitude changes of the cosine waveform and the scanning angle, HPBW, SLL, and maximum scanning range $\theta_{r}$ for 8-element ULAA at compression ratio R=0.7778.

$\mathrm{Cosine}\;\mathrm{Wave}\;\mathrm{Amplitude}\;A_{c}$	$\mathrm{Main}\;\mathrm{Beam}\;\mathrm{Direction}\;\theta_{o}$, in °	SLL (dB)	HPBW in °	Maximum Scanning Range
0	90°	−29.79 dB	12.8°	$-27.64{}^{\circ}\leq \theta_{r}\leq 27.64{}^{\circ}$
0.25 λ	84.02°	−28.9 dB	12.78°	
0.5 λ	78.35°	−27.14 dB	12.88°	
0.75 λ	72.67°	−24.59 dB	12.98°	
λ	67.52°	−21.74 dB	13.06°	
1.25 λ	62.36°	−19.1 dB	13.03°	
1.5 λ	Grating lobe appearance			
1.75 λ				
2 λ				

**Table 12 sensors-23-06557-t012:** The relation between the amplitude changes of the cosine waveform and the scanning angle, HPBW, SLL, and maximum scanning range $\theta_{r}$ for 8-element ULAA at compression ratio R=0.7.

$\mathrm{Cosine}\;\mathrm{Wave}\;\mathrm{Amplitude}\;A_{c}$	$\mathrm{Main}\;\mathrm{Beam}\;\mathrm{Direction}\;\theta_{o}$, in °	SLL (dB)	HPBW in °	Maximum Scanning Range
0	90°	−29.79 dB	12.8°	$-29.19{}^{\circ}\leq \theta_{r}\leq 29.19{}^{\circ}$
0.25 λ	85.05°	−29.28 dB	12.82°	
0.5 λ	79.89°	−28.21 dB	12.95°	
0.75 λ	74.74°	−26.2 dB	13.17°	
λ	70.1°	−23.44 dB	13.42°	
1.25 λ	65.46°	−18.94 dB	13.64°	
1.5 λ	60.81°	−12.68 dB	13.69°	
1.75 λ	Grating lobe appearance			
2 λ				

**Table 13 sensors-23-06557-t013:** The relation between the amplitude changes of the cosine waveform and the scanning angle, HPBW, SLL, and maximum scanning range $\theta_{r}$ for 8-element ULAA at compression ratio R=0.5.

$\mathrm{Cosine}\;\mathrm{Wave}\;\mathrm{Amplitude}\;A_{c}$	$\mathrm{Main}\;\mathrm{Beam}\;\mathrm{Direction}\;\theta_{o}$, in °	SLL (dB)	HPBW in °	Maximum Scanning Range
0	90°	−29.79 dB	12.8°	$-24.03{}^{\circ}\leq \theta_{r}\leq 24.03{}^{\circ}$
0.25 λ	87.11°	−29.77 dB	12.81°	
0.5 λ	84.02°	−29.82 dB	12.94°	
0.75 λ	80.92°	−29.67 dB	13.16°	
λ	77.83°	−29.05 dB	13.48°	
1.25 λ	74.74°	−28.06 dB	13.9°	
1.5 λ	72.16°	−30.35 dB	14.46°	
1.75 λ	69.06°	−28.86 dB	15.18°	
2 λ	65.97°	−27.35 dB	15.82°	

**Table 14 sensors-23-06557-t014:** A comparison between the actual elements’ positions in *Y*-direction compared to the corresponding quantized elements’ positions in *Y*-direction at fixed element spacing in *Z*-direction for the realization of EPP-BSA approach using the proposed PAA thinning.

$\mathrm{Cosine}\;\mathrm{Wave}\;\mathrm{Amplitude}\;A_{c}$	Element Position on Y-Direction/λ	Number of Antenna Element on *Z*-Direction							
		1	2	3	4	5	6	7	8
0 λ	Exact position	0	0	0	0	0	0	0	0
	Quantized position	0	0	0	0	0	0	0	0
0.5 λ	Exact position	0.5	0.46	0.35	0.19	0	−0.19	−0.35	−0.46
	Quantized position	0.5	0.5	0.4	0.2	0	−0.2	−0.4	−0.5
λ	Exact position	1	0.9	0.7	0.4	0	−0.4	−0.7	−0.9
	Quantized position	1	0.9	0.7	0.4	0	−0.4	−0.7	−0.9
2 λ	Exact position	2	1.848	1.414	0.7654	0	−0.7654	−1.414	−1.848
	Quantized position	2	1.8	1.4	0.7	0	−0.7	−1.4	−1.8

**Table 15 sensors-23-06557-t015:** A comparison between the CST simulation results and MATLAB simulation results of the proposed EPP/BSA approach using the proposed PAA thinning and actual EPP/BSA indicating the resultant scan angle, SLL, and HPBW at R=0.875.

$\mathrm{Cosine}\;\mathrm{Wave}\;\mathrm{Amplitude}\;A_{c}$	CST Simulation Results Using PAA Thinning			CST Simulation Results Using Actual EPP/BSA			MATLAB Simulation Results Using Actual EPP/BSA		
	Scan Angle	HPBW	SLL	Scan Angle	HPBW	SLL	Scan Angle	HPBW	SLL
0 λ	0°	12.6°	−13.6 dB	0°	12.3°	−13 dB	0°	12.8°	−12.8 dB
0.5 λ	17°	12.2°	−6.5 dB	17°	12°	−7.5 dB	16.81°	12.4°	−7.88 dB
λ	30°	10.9°	−2.8 dB	31°	11.2°	−4.1 dB	31.76°	11.2°	−4.917 dB
2 λ	50°	8.7°	−1.1 dB	51°	9.1°	−2.1 dB	51.36°	9.2°	−2.668 dB

**Table 16 sensors-23-06557-t016:** Comparison with state-of-the-art works.

Ref.	Beam Steering Range (BSR)	Maximum SLL	HPBW
[19]	Relatively continuous beam steering from −50° to 50° by switching eight different sampling states of the switching mechanism.	Poor SLL performance as the realized SLL of the array is higher than the ideal case due to the phase quantization errors that reaches 22.5° in this 3-bit phase shifter.	Whenever the main beam’s direction changes, the HPBW correspondingly changes.
[20]	Switched beam steering at four distinct angles: (−41°,−12°,+15°,and+48°).	High SLL within the range from−9 dB to−4 dB.	Whenever the main beam’s direction changes, the HPBW correspondingly changes.
[21]	Switched beam steering at four distinct angles: (+10°,−38°,+38°,and−10°).	Good SLL performance that equal (−13 dB,−8 dB,−8 dB,−13 dB,and−6 dB) at the directions (+10°,−38°,+38°,and−10°), respectively.	The HPBW is changed by 4.1° from 23.62° to 27.51° when the main beam is steered from +10° to +38° or from −10° to −38°, respectively.
[22]	Continuous beam steering from −30° to 30°.	Maximum SLL equals−9.2 dB	The HPBW is changed by less than 1°.
[23]	Continuous beam steering from −20° to 20°.	Not available	Not available
[24]	Switched beam steering from −19° to 19°.	Not available	Not available
This work	Continuous beam steering from −51.36°to51.36°.	High SLL=−2.668 dB at the edges of the steering range.	The HPBW is decreased by 3° at the edges of the steering range.
	Continuous beam steering from −30.73° to 30.73°.	Good SLL<−10 dB over the entire steering range.	Constant HPBW over the entire steering range.

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
