# Peer review of "Linear Antenna Array Sectorized Beam Scanning Approaches Using Element Position Perturbation in the Azimuth Plane"

_sensors, 2023, doi:10.3390/s23146557_

Round 1

Reviewer 1 Report

Authors propose two sector beam scanning approaches based on EPP technique denoted as EPP-BSA and SLL/EPP-BSA. It seems that both approaches have different advantages in terms of wide scanning ranges  and high SLLs at the endpoints of the scanning range.

- It is suggested that the authors may add more references related with this work to enhance the literature review for the related topic. Furthermore, the paper may have become in more readable form if they shorthen some sections in the paper.  

- The abbreviation SLL is firstly given in the keywords. SLL should be firstly expressed in the Abstract. 

- Comparison table should be summarized. It seems very long.

- Although there is no implementation in the paper, the authors claim that the proposed approaches provide ease of implementation. It could be better to support their approaches with experimental studies. 

Author Response

Thank you for accepting our manuscript with an opportunity to address the reviewers’ comments.

We are uploading our point-by-point response to the comments (below) 

Reviewer 2 Report

The manuscript presented a perturbation (reshaping) of the location of antenna array elements by changing the position of the array elements perpendicular to the array direction to form a sine wave line distribution and hence scanning the far field pattern. The manuscript is long and continues repeated information in some paragraphs. Modifications are required:

1.     Comments on the main contribution points at the end of the introduction section:

a.     The continuous steering can be achieved in any array antenna configuration, where it mainly depends on the phased array circuit performance. Please clarify this contribution why the proposed antenna is better in this way.

b.     Controlling the array elements' position is more challenging than electronically controlling the progressive phase between the elements. For that, it is not simple to scan the beam. Please clarify the techniques that can be used practically to change this parameter.

c.     Verification of the proposed concept using full-wave simulation software cannot be considered the main contribution since this becomes a must to verify the claims and the mathematical model implemented in numerical software like MATLAB.

2.     All figures in the manuscript should have text and symbol sizes that match and are readable. In addition to that, the legends text is so small.

3.     The case studies in section 2.1.1 are hard to follow. As a recommendation, if possible, you can keep the figure and summarize the discussion by focusing only on the main contribution you want to highlight by changing the compression ratio R.

4.     The proposed antenna utilized a larger area compared to a 1D array. Please elaborate.    

It is acceptable.

Author Response

(The authors gave the same response as above.)

Round 2

Reviewer 2 Report

The authors addressed my comments and made the necessary changes to the manuscript.